EMBO
Molecular Medicine

# Liver TET1 promotes metabolic dysfunction-associated steatotic liver disease

Hongze Chen [1,2,9], Muhammad Azhar Nisar [1,9], Joud Mulla [3,9], Xinjian Li [1,2], Kevin Cao[3], Shaolei Lu[4], Katsuya Nagaoka [3], Shang Wu [1], Peng-Sheng Ting [5], Tung-Sung Tseng [6], Hui-Yi Lin[6], Xiao-Ming Yin[1], Wenke Feng[7], Zhijin Wu [8], Zhixiang Cheng[3], William Mueller[3], Amalia Bay[3], Layla Schechner [1], Xuewei Bai[2,3] & Chiung-Kuei Huang [1✉]

## Abstract

Global hepatic DNA methylation change has been linked to human patients with metabolic dysfunction-associated steatotic liver disease (MASLD). DNA demethylation is regulated by the TET family proteins, whose enzymatic activities require 2-oxoglutarate (2-OG) and iron that both are elevated in human MASLD patients. We aimed to investigate liver TET1 in MASLD progression. Depleting TET1 using two different strategies substantially alleviated MASLD progression. Knockout (KO) of TET1 slightly improved diet induced obesity and glucose homeostasis. Intriguingly, hepatic cholesterols, triglycerides, and CD36 were significantly decreased upon TET1 depletion. Consistently, liver specific TET1 KO led to improvement of MASLD progression. Mechanistically, TET1 promoted CD36 expression through transcriptional upregulation via DNA demethylation control. Overexpression of CD36 reversed the impacts of TET1 downregulation on fatty acid uptake in hepatocytes. More importantly, targeting TET1 with a small molecule inhibitor significantly suppressed MASLD progression. Conclusively, liver TET1 plays a deleterious role in MASLD, suggesting the potential of targeting TET1 in hepatocytes to suppress MASLD.

**Keywords** Fatty Liver; Alcoholic Liver Disease; Nonalcoholic Fatty Liver Disease; Epigenetics; 5-Hydroxymethylcytosine
**Subject Categories** Chromatin, Transcription & Genomics; Digestive System; Metabolism

## Introduction

The metabolic dysfunction-associated steatotic liver disease (MASLD) is the primary type of fatty liver diseases with fat built up in the liver, not caused by alcohol consumption. The global prevalence of MASLD is about 24% (Younossi et al, 2018). MASLD affects 100 million Americans and costs the US healthcare system $103 billion annually, according to a recent analysis of the Medicare database (Sayiner et al, 2017). In certain patients, MASLD can progress to liver fibrosis and further advance to cirrhosis and hepatocellular carcinoma. MASLD affects more than 10–40% in adults worldwide, with 60% in diabetic patients and 90% in obese people. Given that liver cirrhosis is the 12th leading cause of death and liver cancer is the 5th leading cause of cancer-associated death, it is urgent to find a cure for fatty liver diseases. Understanding the underlying mechanisms of MASLD development will likely identify potential therapeutic approaches toward this disease.

It has been recently demonstrated that global DNA methylation is altered in MASLD patients and is highly associated with disease progression (Lai et al, 2020). DNA methylation is involved in adding a methyl group to the fifth carbon pyrimidine ring of cytosine, named as 5-methylcytosine (5mC). The DNA methylation is catalyzed by DNA methyltransferases (DNMTs), including DNMT1, DNMT3a, and DNMT3b (Lyko, 2018). It was previously suggested that DNA demethylation only occurred passively. However, the hydroxymethylcytosine (5hmC) has been recently recognized as the potentially active DNA demethylation mechanism (Guo et al, 2011; Stoyanova et al, 2021). The 5mC can be oxidized by the Ten eleven translocation (TET) family proteins, including TET1, TET2, and TET3, to 5-hmC, 5-formylcytosine (5fC), and 5-carboxylcytosine (5caC). 5fC or 5caC can be excised by the thymine DNA glycosylase (TDG) and repaired by the base excision repair process, thus subsequently resulting in unmodified C. There are limited studies investigating the TET family proteins

[1]Department of Pathology and Laboratory Medicine, Tulane University School of Medicine, New Orleans, LA, USA. [2]Department of Pancreatic and Biliary Surgery, First Affiliated Hospital of Harbin Medical University, 23 Youzheng Street, Nangang District, Harbin 150001 Heilongjiang Province, China. [3]Liver Research Center, Division of Gastroenterology & Liver Research Center, Warren Alpert Medical School of Brown University and Rhode Island Hospital, Providence, RI, USA. [4]Department of Pathology and Laboratory Medicine, Warren Alpert Medical School of Brown University, Rhode Island Hospital, Providence, RI, USA. [5]Department of Medicine, Tulane University School of Medicine, New Orleans, LA, USA. [6]School of Public Health, Louisiana State University Health Sciences Center, New Orleans, USA. [7]Department Structural Cellular Biology, Tulane University School of Medicine, New Orleans, LA, USA. [8]Department of Biostatistics, School of Public Health, Brown University, Providence, RI, USA. [9]These authors contributed equally: Hongze Chen, Muhammad Azhar Nisar, Joud Mulla. ✉E-mail: chuang17@tulane.edu

in MASLD development. A clinical study investigating 5hmC in MASLD revealed that there is no difference in nuclear 5hmC between normal and MASLD patients. However, MASLD patients have significantly lower non-nuclear 5hmC signals than the control ones. Furthermore, they identified that TET1 and TET2 missense variants (loss of function) are potentially associated with MASLD severity (Pirola et al, 2015). Interestingly, 5hmC and all three TET enzymes have been found either downregulated or no change in human liver tissue samples with chronic liver diseases (Ji et al, 2019; Page et al, 2016). In contrast, the expression levels of DNMT1, DNMT3a, and DNMT3b were found upregulated in human fibrotic liver tissues (Page et al, 2016). In line with the human findings, it has been recently reported that DNMT1 and DNMT3a increased in male C57BL/6J mice fed with a high fat diet. Liver specific DNMT1 or DNMT3a knockout (KO), which resulted in DNA hypomethylation, both led to alleviation of MASLD progression (Wang et al, 2023). Besides, they found that liver-specific TET2 KO exacerbated MASLD. Interestingly, it was previously reported that liver specific KO of TET2 and TET3 elicited fatty liver development spontaneously (Reizel et al, 2018). TET2 and TET3 KO would lead to reduced 5hmC formation, resulting in DNA hypermethylation. However, all these findings could not fully explain the clinical observations that DNA methylation is decreased in MASLD patients and negatively correlated with disease severity. Intriguingly, both 2-oxoglutarate and iron, which are required for the TETs' enzymatic activity, are elevated in MASLD patients (Dongiovanni et al, 2011; Rodriguez-Gallego et al, 2015), suggesting that downregulated DNA methylation in human MASLD patients likely due to the elevated enzymatic activity of TETs. However, liver TET2 and TET3 KO cannot protect mice from fatty liver development. Thus, we speculated that elevated liver TET1 function promotes MASLD progression and aimed to determine the role of liver TET1 in MASLD development.

The current study adopted two whole body TET1 KO strains and liver specific TET1 KO mice to investigate the role of TET1 in MASLD. We clarify that hepatic TET1 regulates lipid metabolism via transcriptional regulation of CD36. Targeting TET1 with a small molecule inhibitor improved metabolic indexes in mice fed with an HFD, thereby providing the preclinical evidence that targeting liver TET1 is a potential therapeutic approach in patients with MASLD.

# Results

## Profiling DNA methylation/demethylation enzymes in MASLD samples revealed liver TET1 and TET3 are downregulated in MASLD

To determine the global DNA methylation change in mouse liver samples with MASLD, we fed male C57BL/6J mice with an HFD. Phenotype characterization validated the HFD-treated mice with increased body weight (BW), the ratio of liver weight (LW) over BW, and the ratio of fat weight (FW) over BW (Fig. 1A). Besides, obesity, hepatomegaly, and hepatic steatosis were found in these mice (Fig. 1B,C). The global DNA methylation was found elevated in the HFD-treated mice (Fig. 1D,E). To determine how hepatic global DNA methylation is affected, we examined the expression

levels of DNMTs in MASLD patients. We retrieved the RNA-Seq data of GSE174478 and analyzed DNMTs levels in control (CTRL) and MASLD patients. The expression levels DNMT1 and DNMT3a were not significantly different between human CTRL and MASLD groups. However, DNMT3b is increased in the human MASLD group (Appendix Fig. S1A–C). We also examined DNMTs in preclinical mouse MASLD samples. It was noted that DNMT1, DNMT3a, and DNMT3b were not significantly altered in the HFD fed mouse group despite that DNMT1 was slightly increased in the mouse MASLD group (Appendix Fig. S1D–F). We further examined the expression levels of TET1, TET2, and TET3 in these samples. Similarly, no significant changes of TET1, TET2, and TET3 were found between human/mouse CTRL and MASLD groups (Appendix Fig. S2A–F). A previous study has illustrated the involvement of DNMT1 and DNMT3a in regulating MASLD development (Wang et al, 2023). To comprehensively understand the DNA methylation control in MASLD development, we therefore examined the TET family proteins in these mouse MASLD samples. Unexpectedly, we found that TET1 and TET3 protein were significantly decreased in the liver samples of MASLD mice (Fig. 1F), indicating the potential impacts of TET1 and TET3 on MASLD development.

## Whole body TET1 KO suppresses MASLD progression

Given the previous finding that TET2 and TET3 liver specific KO promoted fatty liver development in mice, we decided to investigate the role of TET1 in hepatic steatosis (Reizel et al, 2018). We generated and characterized the TET1 whole body KO mice. TET1 deletion genotyping was verified (Appendix Fig. S3A). When we examined LW/BW, we did not observe any significant difference between control and TET1 KO mice, indicating that TET1 deletion has minimal impacts on the liver development (Appendix Fig. S3B). Interestingly, in line with the previous study that adipocyte-specific TET1 KO would lead to reduced FW/BW in mice fed with either normal chow (NC) or HFD (Damal Villivalam et al, 2020), the TET1 KO mice also have a slight decrease in FW/BW when compared with the control ones (Appendix Fig. S3C). We also checked the BW of the littermate control and TET1 KO mice starting from 8 weeks weekly. The results suggested that TET1 KO mice may have slightly lower BW than the control ones (Appendix Fig. S3D). We further examined if liver histologic development is affected by TET1 KO via evaluating the hepatic histology. As shown, TET1 KO did not significantly impact liver histological development (Appendix Fig. S3E). Thereafter, these male mice were fed with an HFD for 16 weeks. It was found that TET1 KO reduced mouse BW compared to the littermate controls (Fig. 2A). TET1 KO mice have healthy liver appearance, unlike control mice having the yellow-brown one (Fig. 2B). The LW/BW was found decreased in TET1 KO mice (Fig. 2C). Unexpectedly, TET1 KO mice have a higher FW/BW than the control ones (Fig. 2D). Nevertheless, aspartate aminotransferase (AST) and alanine transaminase (ALT) enzymatic activity assays, the liver functional studies demonstrated that TET1 KO substantially alleviated liver damages in the HFD-induced MASLD progression (Fig. 2E,F). These findings were reproducible using another strain of TET1 KO mice generated by crossing the floxed TET1 mice (Kang et al, 2015) with β-actin promoter driven Cre recombinase ones (Fig. 2G–I). These results demonstrated a deleterious role of TET1 in MASLD development.

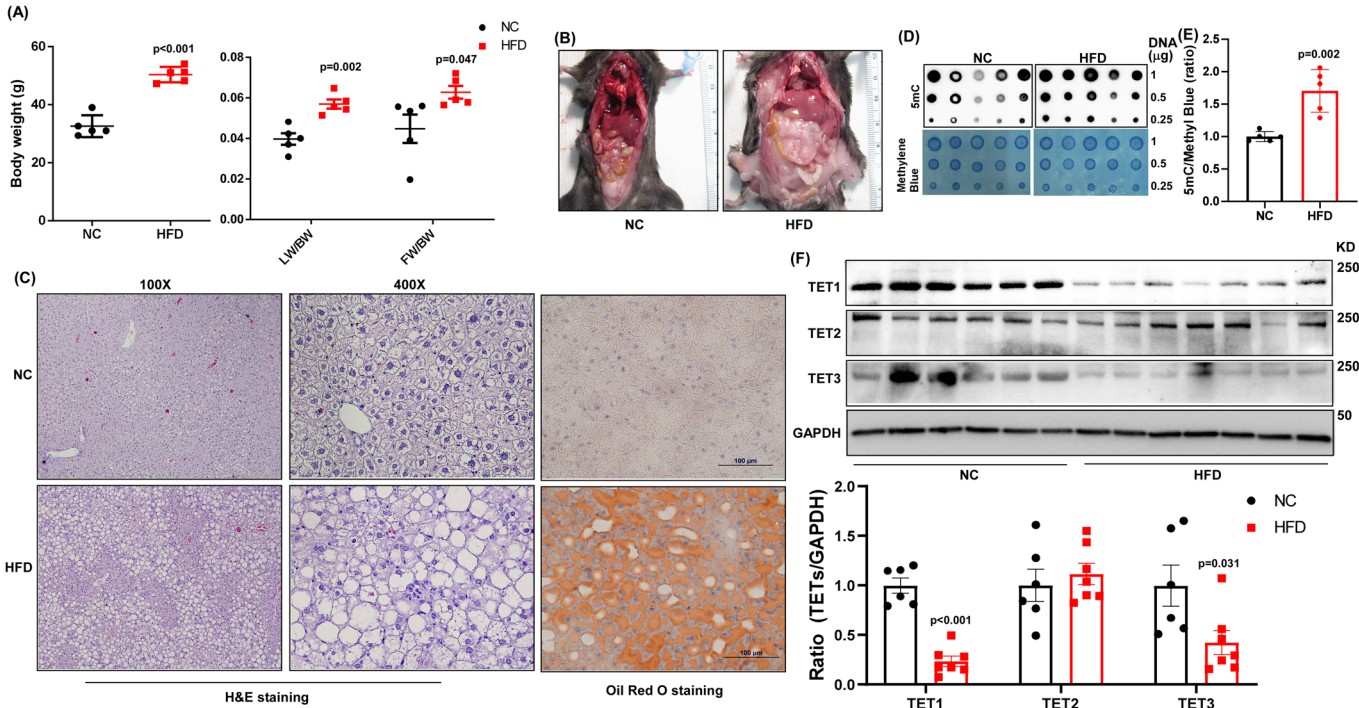

**Figure 1. Hepatic TET1 and TET3 are downregulated in MASLD.**

(A) Body weight, liver weight over body weight (LW/BW), and visceral fat weight over BW (FW/BW) were examined in male C57/BL6 mice fed with either a normal chow (NC) or high-fat diet (HFD) for 16 weeks, $n = 5$ in NC; $n = 5$ in HFD. The representative (B) gross images, (C) 100X H&E, 400X H&E, and 400X Oil Red O staining images of NC and HFD fed mice. (D) 5mC dot blot and methylene blue staining were done using the genomic DNA of liver samples derived from NC and HFD mice. (E) Quantification results of 5mC/methylene blue, $n = 5$ in NC; $n = 5$ in HFD (F) TET1, TET2, and TET3 protein expression levels were determined in NC and HFD treated mice. Bottom panel: The relative ratios of TET1, TET2, TET3 over GAPDH, $n = 6$ in NC; $n = 7$ in HFD Data are presented as mean ± S.D. in (A, E), and (F). P values were determined using unpaired two-tailed Student's T test. Exact p values were as indicated. Source data are available online for this figure.

## TET1 KO improved diet induced obesity and glucose homeostasis

We have found that TET1 KO suppressed diet induced obesity and MASLD (Fig. 2A–C,H). To illustrate the mechanisms by which TET1 affects MASLD, we measured food and water consumption in control and TET1 KO mice fed with an HFD given that the previous studies have demonstrated an important role of TET1 in neuron development (Zhang et al, 2013) which would possibly impact food intake due to disrupted appetite (Qi et al, 2023). Although TET1 is involved in neuron development, the overall impacts of TET1 KO on food and water consumption have not been significantly affected (Appendix Fig. S4A,B) by the TET1-mediated neurogenesis. In line with the previous study (Zhang et al, 2021), fed glucose levels were not significantly changed in TET1 KO mice (Appendix Fig. S4C,D). Interestingly, insulin levels were found decreased in TET1 KO mice (Appendix Fig. S4E). As a previous study demonstrated the involvement of AMPK-TET1-Sirt1 axis in glucose metabolism through regulating GLUT4 and PGC1α (Zhang et al, 2021), we also examined hepatic GLUT4 and PGC1a in control and TET1 KO mice fed with an HFD. We observed no significant difference of these genes between control and TET1 KO mice (Appendix Fig. S4F). We also examined the insulin response indicator, phosphorylated Akt (pAkt), but found no significant change. Given the potential involvement of AMPK signaling in TET1-mediated glucose metabolism, we further

examined the AMPK signaling pathway. The AMPK upstream kinase, LKB1 and pLKB1 were not affected by TET1 KO. Neither pAMPK nor AMPK was impacted by TET1 KO (Appendix Fig. S4G). An unbiased mRNA sequencing experiment was performed using the liver samples derived from control and TET1 KO mice fed with an HFD. Several hepatokines, including Eda, Enho, Hsp90aa1, Igfbp2, Igfbp3, and Slc27a1 which have been linked to MASLD were found significantly upregulated in TET1 KO mice (Dataset EV1, EV2). Further examining these genes in TET1 liver specific KO (LKO) mice exhibited that only Enho is increased in TET1 LKO mice (Appendix Fig. S5). Given the facts that Enho overexpression improves MASLD glucose homeostasis, that TET1 heterozygous KO improved glucose and insulin tolerance tests, and that TET1 KO reduced serum insulin levels, TET1 KO could likely suppress MASLD progression in part through improving glucose homeostasis.

## TET1 KO repressed MASLD likely through inhibiting hepatic lipid accumulation

The liver plays a crucial role in lipid metabolism, functioning as central organ in fatty acid metabolism. Thus, we examined lipid accumulation in the liver of control and TET1 KO mice. As shown, significant amounts of lipids were found in control mice fed with an HFD but not in TET1 KO mice (Fig. 3A). Further analyzing lipid profiles in the liver tissues revealed that TET1 KO decreased

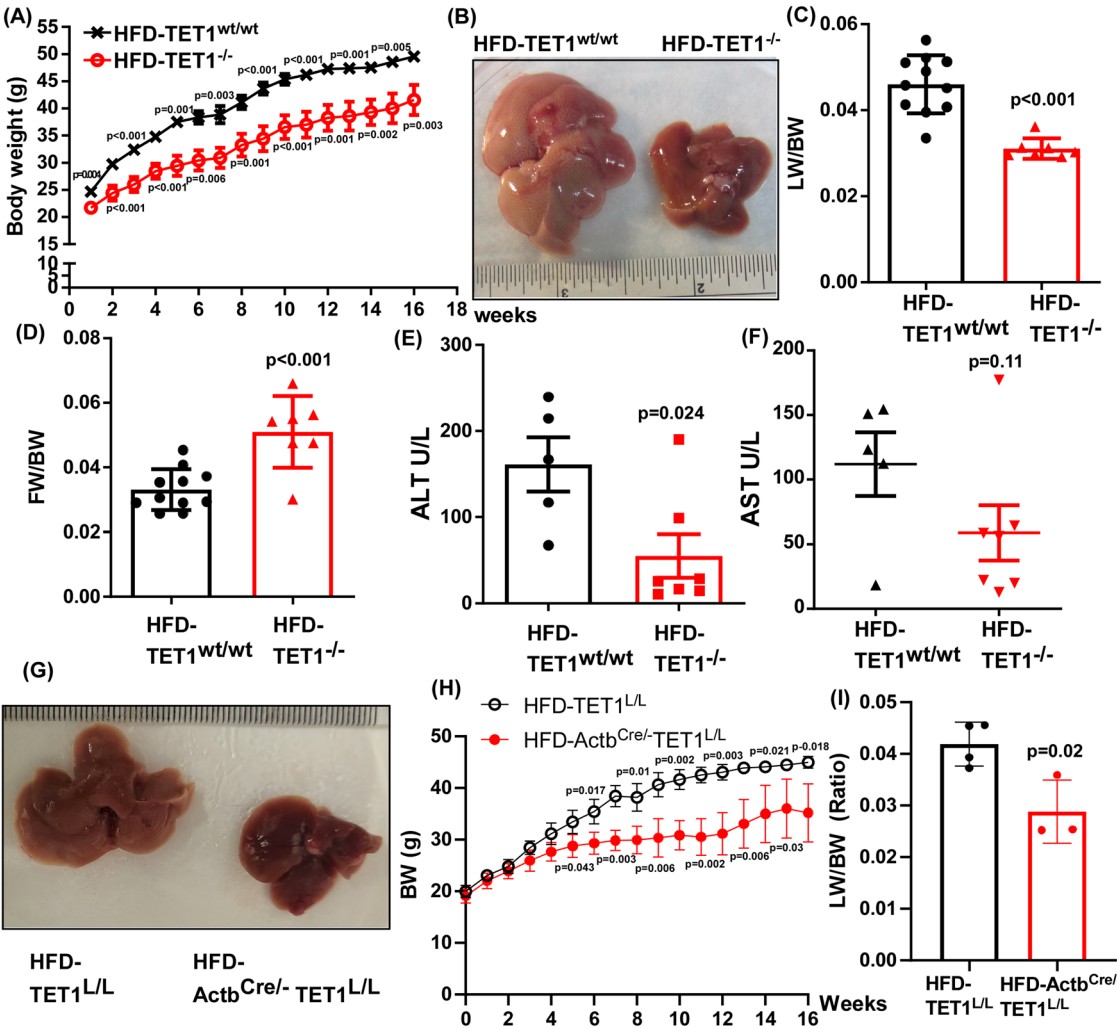

**Figure 2. Ubiquitously knocking out TET1 using two strategies suppressed MASLD progression.**

(A) Body weight was measured in male TET1^wt/wt^ and TET1^−/−^ mice fed with an HFD for 16 weeks starting at 8 weeks old, n = 11 in TET1^wt/wt^; n = 7 in TET1^−/−^. (B) The representative liver images of TET1^wt/wt^ and TET1^−/−^ mice fed with an HFD. (C) LW/BW, (D) FW/BW, (E) ALT, and (F) AST were determined in these mice. For (C) and (D), n = 11 in TET1^wt/wt^; n = 7 in TET1^−/−^. For (E) and (F), n = 5 in TET1^wt/wt^; n = 7 in TET1^−/−^. (G) The gross liver images of male TET1^L/L^ and Actb^Cre/-^ TET1^L/L^ mice fed with an HFD for 16 weeks starting at 8 weeks old. (H) BW and (I) LW/BW were determined in these mice, n = 4 in TET1^L/L^; n = 3 in Actb^Cre/-^TET1^L/L^ Data are presented as mean ± S.D. in (A, C–F, H), and (I). P values were determined using unpaired two-tailed Student's T test. Exact p values were as indicated. Source data are available online for this figure.

hepatic cholesterol and triglyceride levels in mice fed with an HFD (Fig. 3B,C). The lipidomic assay and enrichment analysis of lipidomic data, performed using Lipid Pathway Enrichment Analysis (LIPEA) and Kyoto Encyclopedia of Genes and Genomes (KEGG) tools, reveals that the most significant altered pathway in the TET1 LKO and TET1 KO groups compared to the control one is glycerophospholipid metabolism The key upregulated lipids in this pathway are 1-Acyl-sn-glycero-3-phosphocholine, 2-Acyl-sn-glycero-3-phosphocholine, 1-Acyl-sn-glycero-3-phosphoethanolamine, and 2-Acyl-sn-glycero-3-phosphoethanolamine. This leads to the significant enrichment of both the phosphatidylethanolamine and phosphatidylcholine biosynthesis pathways. Conversely, the downregulated lipids in the glycerophospholipid metabolism pathway are 1-Acyl-sn-glycerol-3-phosphate and 2-Acyl-sn-glycerol-3-phosphate (Dataset EV3, 4, Table EV1 and Appendix Fig. S6A,B).

These phenotype changes are likely mediated by de novo lipogenesis (DNL), fatty acid β-oxidation, fatty acid uptake, and very low density lipoprotein (VLDL) synthesis and secretion mechanisms (Alves-Bezerra and Cohen, 2017; Bechmann et al, 2012; Ipsen et al, 2018; Kawano and Cohen, 2013; Nguyen et al, 2008). As the control and TET1 KO mice ate equal amounts of foods, they should have equal circulating cholesterols and triglycerides if there is no difference in lipid metabolism in the liver samples of these mice. Although the circulating triglyceride levels were not affected by TET1 KO, the circulating cholesterols, hepatic cholesterols, and hepatic triglycerides were significantly reduced in TET1 KO mice (Fig. 3B–E), indicating the involvement of TET1 in hepatic lipid metabolism. We then analyzed differentially expressed genes (DEGs) and revealed that 265 upregulated and 369 downregulated genes were identified in

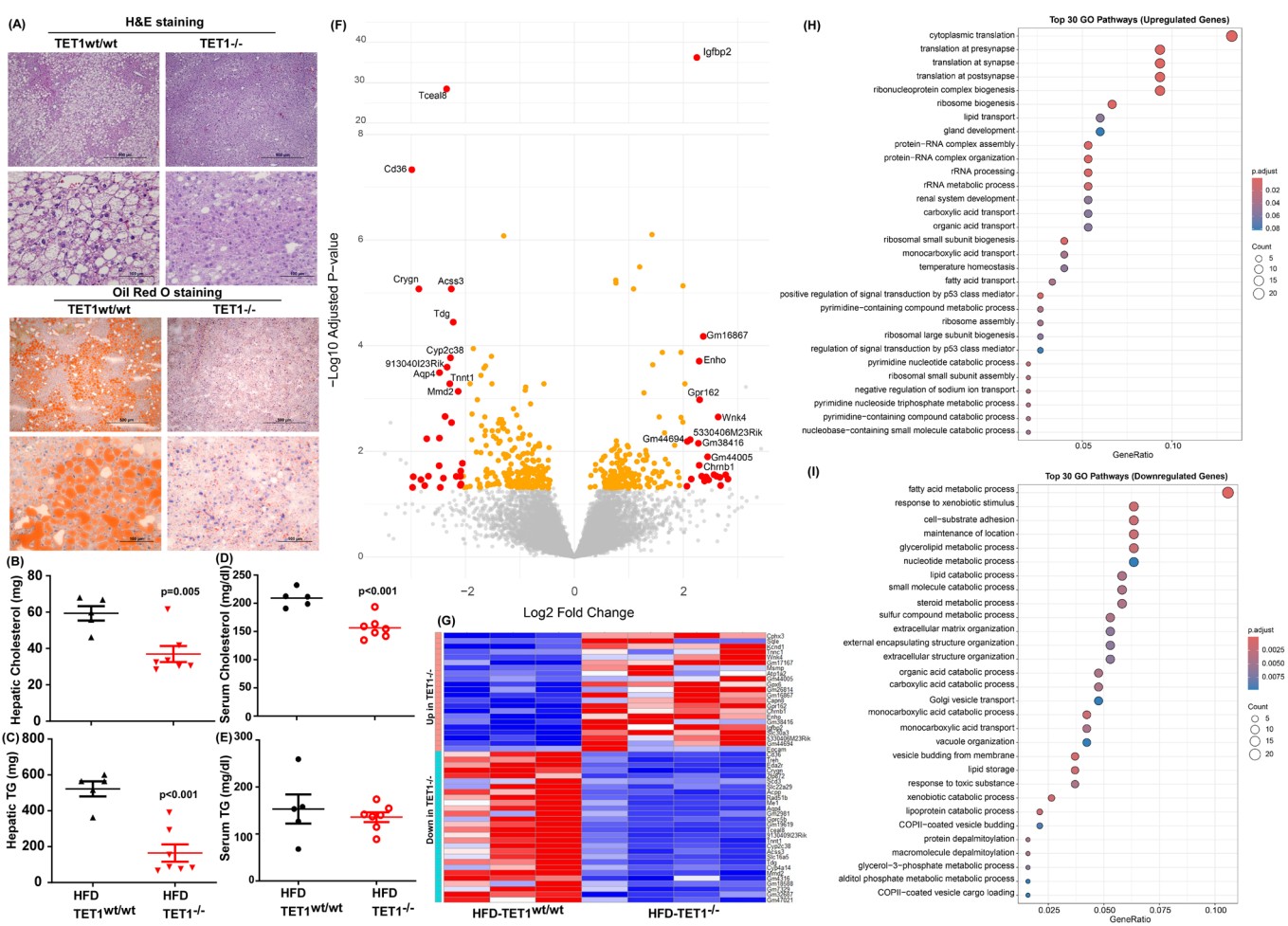

**Figure 3.  TET1 KO suppressed hepatic lipid contents through targeting hepatic lipid metabolism.**

(A) The representative H&E and Oil Red O staining images of TET1$^{wt/wt}$ and TET1$^{-/-}$ mice suggested decreased lipid deposition in TET1 KO mice. (B) Hepatic cholesterols, (C) hepatic triglycerides (TG), (D) serum cholesterols, and (E) serum TG were measured in male TET1$^{wt/wt}$ and TET1$^{-/-}$ mice fed with an HFD, $n = 5$ in TET1$^{wt/wt}$; $n = 7$ in TET1$^{-/-}$ By using RNA sequencing, hepatic differentially expressed genes (TET1$^{-/-}$ versus TET1$^{wt/wt}$) were determined in the liver samples derived from male TET1$^{wt/wt}$ and TET1$^{-/-}$ mice fed with an HFD for 16 weeks. (F) A volcano plot of DEGs exhibits significantly downregulated genes which include CD36 in TET1 KO mice on the left-hand side. (G) Top altered 50 genes including CD36 and Igfbp2 were listed in a heat map. The gene set enrichment analysis using GO term revealed significantly (H) upregulated and (I) downregulated pathways in TET1$^{wt/wt}$ mice. Pathways involved in lipid metabolism were highly downregulated in TET1$^{-/-}$ mice. Data are presented as mean ± S.D. in (B–D), and (E). P values were determined using unpaired two-tailed Student's T test. Exact *p* values were as indicated. Source data are available online for this figure.

TET1 KO mice compared to the control ones (Table EV1 and Fig. 3F), among which top 50 altered genes are illustrated (Fig. 3G). Next, the analysis using the gene set enrichment analysis approach with the gene ontology sets showed that lipid metabolic process, fatty acid metabolic process, and cellular lipid metabolic process are three of the major pathways downregulated in TET1 KO mice (Fig. 3H,I). Collectively, TET1 KO alleviated MASLD development likely through inhibiting the expression of genes associated with lipid metabolism.

## TET1 promotes free fatty acid uptake in hepatocytes by upregulating CD36

Among the most significantly affected top 10 DEGs, CD36 and IGFBP2 are highly associated with MASLD development (Hedbacker et al, 2010; Wilson et al, 2016) (Fig. 4A,B). To validate if

TET1 modulates these genes in hepatocytes directly, we knocked down TET1 in human hepatocytes using two different TET1 shRNAs. Decreased TET1 mRNA and protein expression levels were confirmed in hepatocytes treated with TET1 shRNAs (Fig. 4C,D). Besides, the functional impacts of TET1 knockdown on 5hmC were validated as 5hmC levels were reduced in TET1 knockdown hepatocytes (Fig. 4E,F). Consistently, TET1 deficiency promoted IGFBP2 but suppressed CD36 expression (Fig. 4G,H). Knockdown of TET1 reduced oleic acid uptake in human hepatocytes as evidenced with reduced lipid content using the Oil Red O staining (Fig. 4I,J). To further validate the impacts of TET1 on fatty acid uptake, we challenged these hepatocytes with a fatty acid-conjugated fluorescent. TET1 deficiency suppressed fatty acid uptake since TET1 knockdown reduced fluorescence intensity (Fig. 4K,L). In line with these findings, activating TET1 with its agonist-2-oxoglutarate enhanced fatty acid uptake; whereas

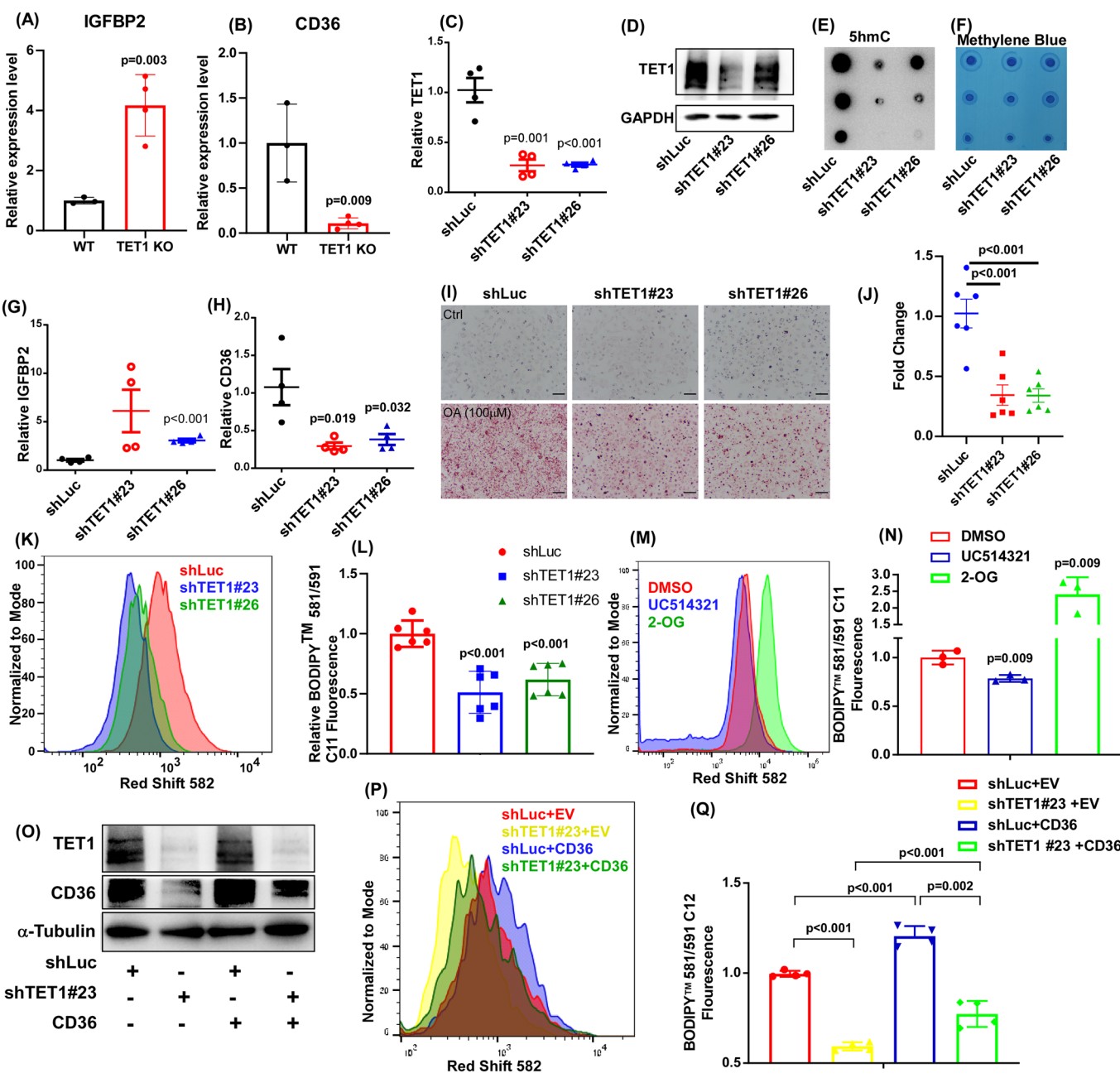

**Figure 4. CD36 is involved in the TET1-mediated free fatty acid uptake.**

(A) IGFBP2 and (B) CD36 genes were examined in the liver samples of TET1^{wt/wt} and TET1^{−/−} mice fed with an HFD, $n = 3$ in WT; $n = 4$ in TET1 KO for (A) and (B, C) TET1 mRNA, (D) protein, (E) 5hmC formation, (F) methylene blue, (G) IGFBP2, (H) CD36 were determined in shRNA-luciferase (shLuc), shRNA-TET1#23 (shTET1#23), and shRNA-TET1#26 (shTET1#26) treated human hepatocytes, $n = 4$ in shLuc, shTET1#23, and shTET1#26 for (C, G), and (H). (I) The representative Oil Red O staining images of shLuc, shTET1#23, and shTET1#26 human hepatocytes treated with 100 μM oleic acid (OA) for 24 h. All scale bar indicate 50 μM. (J) Semi-quantification results of Oil Red O staining, $n = 6$ for all groups in (J, K) The representative flow cytometry images of free fatty acid uptake were shown for shLuc, shTET1#23, and shTET1#26 human hepatocytes. (L) Quantification results of (K), $n = 6$ for all groups in (L, M) The representative images of free fatty acid uptake in human hepatocytes treated with 200 μM 2-oxoglutarate or 5 μM UC514321 for 48 h. (N) Quantification results of (M), $n = 3$ for all groups in (N, O) The protein expression levels of TET1, CD36, and α-tubulin were determined in human hepatocytes treated with shLuc + empty vector, shLuc + CD36, shTET1#23 + empty vector, shTET1#23 + CD36. (P) The representative images and (Q) quantification results of free fatty acid uptake were shown for these cells, $n = 4$ for all groups in (Q) Data are presented as mean ± S.D. in (A, B, C, G, H, J, L, N), and (Q). P values were determined using unpaired two-tailed Student's T test. Exact p values were as indicated. Source data are available online for this figure.

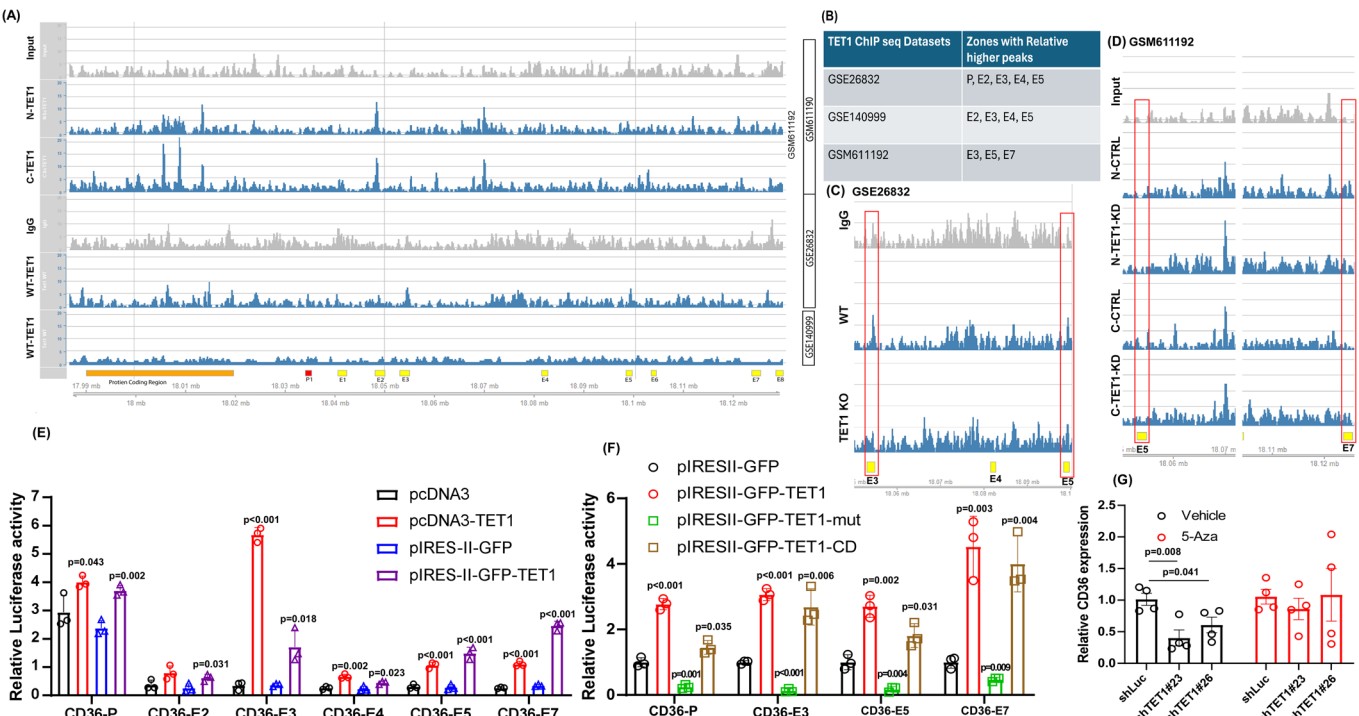

**Figure 5. CD36 is transcriptionally regulated by TET1 through DNA methylation control.**

(A) The representative histograms of TET1 ChIP bindings to the CD36 gene in three datasets, including GSE26832, GSE140999, and GSM611192. (B) Summary of the TET1 binding sites on CD36 promoter (P) and enhancers. (C) The TET1 binding peaks were reduced in enhancer 3 (E3) and E5 of TET1 KO samples derived from the GSE26832 dataset. (D) The TET1 binding peaks of E5 and E7 were decreased in TET1 knockdown (KD) cells compared to the control (CTRL) ones derived from the GSM611192 dataset. (E) The relative luciferase activities of CD36-P, E2, E3, E4, E5, and E7 were determined in human cells treated as indicated for 24 h, $n = 3$ for all groups (F) Relative luciferase activities of CD36-P, E3, E5, and E7 were examined in human cells transfected with pIRESII-GFP control vector, pIRESII-GFP-TET1, pIRESII-GFP-TET1-mut (with catalytic domain mutation), pIRESII-GFP-TET1-CD (with TET1 catalytic domain only) for 24 h, $n = 3$ for all groups. (G) Relative CD36 mRNA expression levels were determined in human hepatocytes manipulated with shLuc, shTET1#23, and shTET1#26 in the presence of DMSO or 10 μM 5-Azacytidine (5-Aza) for 48 h, $n = 4$ for all groups. Data are presented as mean ± S.D. in (E, F), and (G). P values were determined using unpaired two-tailed Student's T test. Exact p values were as indicated. Source data are available online for this figure.

targeting TET1 with the TET1 inhibitor-UC514321 suppressed it (Fig. 4M,N). Although both IGFBP2 and CD36 are involved in MASLD, the major function of CD36 is highly involved in fatty acid uptake (Pepino et al, 2014). Besides, recent studies have suggested the positive correlation of IGFBP2 and advanced MASLD, fibrosis (Luo et al, 2021; Sveinbjornsson et al, 2022). Under these circumstances, we decided to investigate if the TET1-mediated fatty acid uptake is through CD36 and how TET1 modulates CD36. We re-expressed CD36 in TET1 knockdown human hepatocytes (Fig. 4O) and repeated the fatty acid uptake experiments. As shown, re-expressing CD36 in TET1 knockdown hepatocytes significantly rescued the impact of TET1 deficiency on fatty acid uptake (Fig. 4P,Q). These findings suggested that the TET1-mediated CD36 in hepatocytes plays a boosting role in the upregulation of fatty acid uptake, likely contributing to MASLD progression.

## CD36 is regulated by the TET1-mediated DNA demethylation control

It has been previously suggested that the transcriptional control of CD36 is regulated by the promoters and several enhancers (Mikkelsen et al, 2010). To determine whether TET1 promotes CD36 through affecting transcriptional regulation, we specifically

investigated TET1 binding to these regions by retrieving three TET1 chromatin immunoprecipitation sequencing (ChIP-Seq) datasets using mouse samples, including GSE140999, GSE26832, and GSM611192. It was found that TET1 bound to the protein coding region, promoter region (P), enhancer 2 (E2), E3, E4, E5, and E7 regions, when compared the results of wildtype (WT) mice to the input positive and IgG negative controls (Fig. 5A,B). To further exclude TET1 false positive bindings, we analyzed the ChIP-Seq data (GSE26832) derived from WT and TET1 KO mice. We also analyzed the ChIP-Seq data (GSM611192) generated using N and C terminal antibodies in TET1-control (CTRL) and TET1-knockdown (TET1 KD) treated mouse cells. We found that the TET1 binding peaks at E3 and E5 regions are reduced in TET1 KO mice. Besides, TET1 binding peaks at E5 and E7 were decreased in TET1 KD cells compared to the CTRL ones (Fig. 5C,D). These findings suggested the importance of P, E2, E3, E4, E5, and E7 in TET1-mediated CD36 regulation. In addition, TET1 likely has more impacts on the E3, E5, and E7 regulation on CD36 transcription than the other regions. Thus, we further investigated whether overexpression of TET1 enhances the transcriptional activation of CD36 P, E2, E3, E4, E5, and E7 by using two sets of TET1 overexpression systems (pcDNA3-TET1 from mouse and pIRES-II-GFP-TET1 from human). Indeed, overexpression of

TET1 promoted the transactivation of CD36 P, E3, E4, E5, and E7 (Fig. 5E). Consistent to the ChIP-Seq analysis, overexpression of TET1 specifically and robustly enhanced the transactivation of CD36 E3, E5, and E7. To further determine if these regulations are dependent on TET1 catalytic activity, we overexpressed vector control, TET1, TET1 with catalytic domain dead mutation (TET1-Mut), and TET1 catalytic domain only (TET1-CD) in combination with CD36 P, E3, E5, and E7-luciferaase. Interestingly, both TET1 and TET1-CD could promote CD36 P, E3, E5, and E7 transactivation but not TET1-mut (Fig. 5F). Further, treating human hepatocytes with the DNA methylation inhibitor-5-Azacytidine (5-Aza) could reverse the suppressive effects of TET1 knockdown on CD36 expression (Fig. 5G). These results demonstrated that TET1 upregulates CD36 expression through enhancing transcriptional activation control likely via promoting DNA demethylation.

## Depleting hepatic TET1 alleviates MASLD progression

To further determine the impact of hepatic TET1 on MASLD progression, we generated a novel mouse strain of TET1 LKO by breeding the floxed TET1 mice with albumin promoter driven Cre ones. As shown, depleting TET1 in the liver resulted in reduced 5hmC formation, but it did not significantly affect mouse BW compared to the littermate controls (Appendix Fig. S7A,B). Neither LW/BW nor liver functions were impacted by specifically knocking out hepatic TET1 in the mice fed with an NC diet (Appendix Fig. S7C,D). Interestingly, deleting hepatic TET1 reduced hepatic cholesterols and triglycerides without significantly affecting serum cholesterols and triglycerides in these mice fed with an NC diet (Appendix Fig. S7E–H). Given these findings, we then challenged control and TET1-LKO mice with an HFD for 16 weeks. Similar to the whole body TET1 KO mice, knocking out hepatic TET1 significantly inhibited BW, LW, LW/BW, but not food consumption, glucose, FW, and FW/BW in mice fed with an HFD. Interestingly, insulin is slightly reduced in TET1 LKO mice (Fig. 6A–D; Appendix Fig. S8A–E). The ALT liver functional assay suggested that hepatic TET1 deletion alleviates MASLD progression (Fig. 6E). Histological and biochemical results show that hepatic and serum cholesterols were reduced in TET1-LKO mice (Fig. 6F–J). Interestingly, hepatic TET1 deletion only suppressed CD36 expression but had no significant impacts on IGFBP2 in mice fed with an HFD (Fig. 6K), further supporting that the TET1-mediated CD36 expression is in hepatocytes. As one of the major CD36 functions is involved in free fatty acid uptake and it has been previously suggested that Pparα and HNF4α act as free fatty acid sensors in the liver (Fougerat et al, 2022; Simcox et al, 2017), we also examined Pparα, Hnf4α, and their downstream target genes. In line with the previous studies, several Pparα and Hnf4a downstream targets were decreased in TET1-LKO mice (Appendix Fig. S9A,B). The expression profiles of TETs also excluded the compensated effects of TET2 and TET3 in hepatic TET1 deletion mice (Fig. 6L). Therefore, these findings indicate the deleterious role of hepatic TET1 in MASLD progression.

## Targeting TET1 leads to improvement of steatosis in mice fed with an HFD

To investigate if targeting TET1 with a small molecule inhibitor can alleviate MASLD progression, we challenged MASLD mice with a

TET1 inhibitor (TET1 i) (Jiang et al, 2017), UC514321, for 8 weeks (Fig. 7A). The TET1 i treatment significantly reduced BW gain, LW, and LW/BW in MASLD mice compared to the control group treated with vehicle solution (Fig. 7B–D). However, FW and FW/BW were not significantly changed between these two groups (Fig. 7E,F). Histological analysis revealed decreased hepatic steatosis in the TET1 i treated group (Fig. 7G). The results of serum and hepatic lipid profiles demonstrated that TET1 i treatment downregulated serum and hepatic TG levels in MASLD mice (Fig. 7H–K). Besides, TET1 i suppressed CD36 mRNA expression in MASLD mice (Fig. 7L). Furthermore, TET1 i significantly inhibited TET1 and CD36 protein expression levels (Fig. 7M,N), strongly supporting the TET1 i-mediated CD36 downregulation contributing to the improved MASLD phenotypes. Taken together, targeting TET1 with a small molecule inhibitor suppressed MASLD progression by reducing lipid deposition in the liver through downregulating CD36 expression.

# Discussion

We elucidated for the first time that liver TET1 plays a deleterious role in MASLD progression. Moreover, we authenticated the impacts of whole body TET1 KO on suppressing MASLD progression using two different strategies. We also revealed the novel regulation of TET1 in free fatty acid uptake by transcriptionally upregulating CD36 through DNA demethylation control. More importantly, we determined that targeting TET1 with a small molecule inhibitor alleviated MASLD progression.

It has been previously demonstrated that cytoplasmic 5hmC is decreased in human MASLD samples (Pirola et al, 2015). Under these circumstances, DNA methylation should be elevated in MASLD patients as we observed in the preclinical MASLD model. Unexpectedly, liver 5mC levels are decreased in human MASLD and correlated with disease progression (Lai et al, 2020). This discrepancy is likely due to elevated alpha-ketoglutarate in human MASLD patients (Rodriguez-Gallego et al, 2015). Interestingly, the study aiming at clarifying post-natal DNA methylation change revealed that inducing hepatic TET2 and TET3 double KO post-natal day 1 would result in MASLD spontaneously. But, these changes were not observed in those mice with hepatic TET2 and TET3 double KO induced post-natal week 3 (Reizel et al, 2018). Consistent with the previous findings, the protective role of TET2 in MASLD has been recently validated (Wang et al, 2023). Although the study also performed TET1 and TET3 knockdown experiments in mice, the authors focused on hepatic TET2 knockdown on MASLD given the robust increase of triglycerides in TET2 knockdown mice (Wang et al, 2023). More recently, a study investigated the role of TET3 in MASLD progression and suggested its harmful role in MASLD associated fibrosis (Sun et al, 2024). Indeed, the deleterious role of TET3 in liver fibrosis has been suggested before (Xu et al, 2020). For the impacts of TET1 on MASLD progression, it has been suggested that depleting adipocyte TET1 protects mice from MASLD through suppressing obesity and insulin resistance (Damal Villivalam et al, 2020). The role of hepatic TET1 has been studied in glucose homeostasis by using whole body TET1 heterozygous KO mice. It was found that TET1 heterozygous deletion improved glucose and insulin tolerance despite that TET1 deficiency disrupted glucose metabolism (Zhang

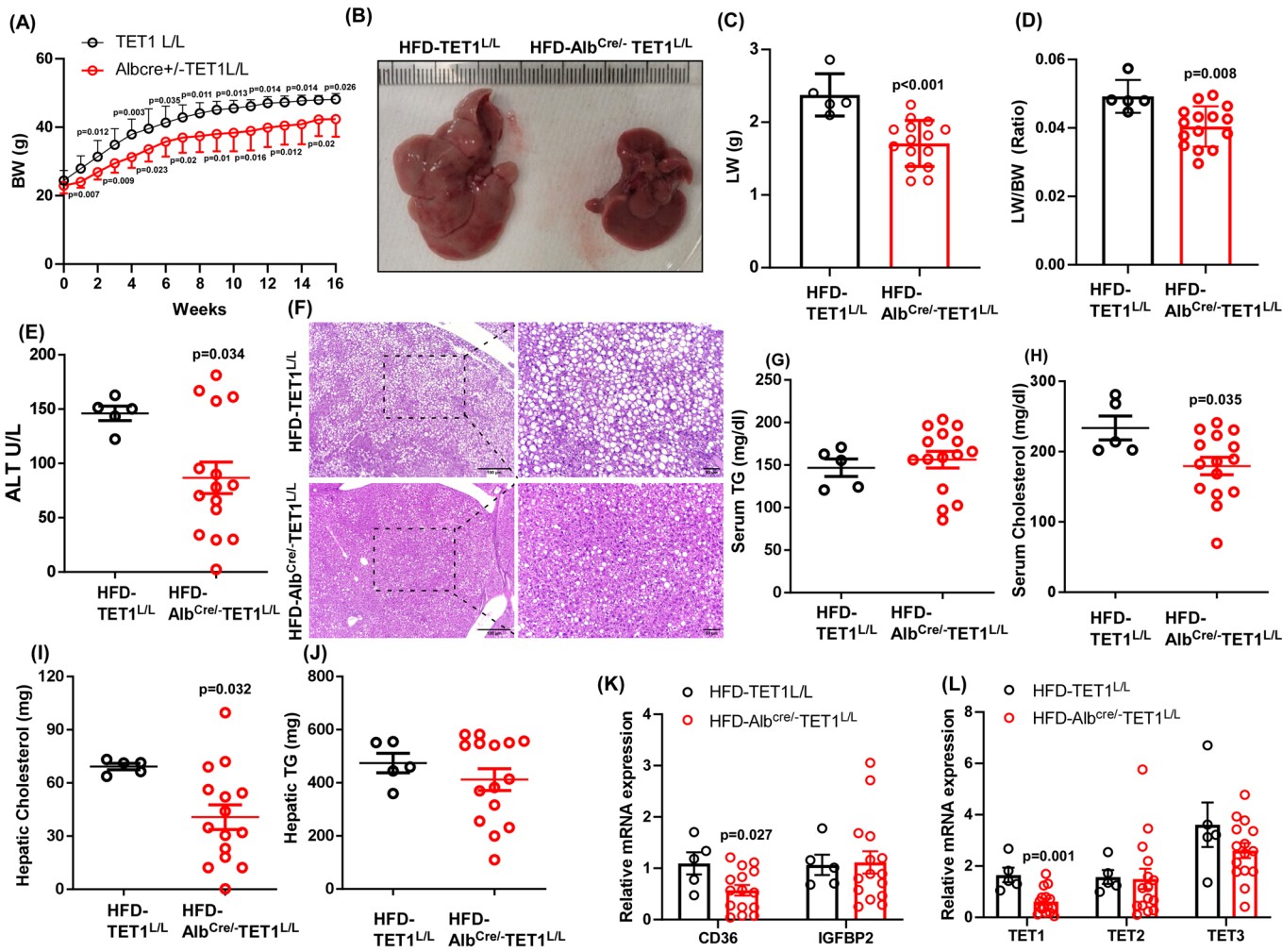

**Figure 6. Liver specific KO inhibited MASLD progression.**

(A) The weekly BW of male TET1[L/L] and Alb[Cre/-] TET1[L/L] mice fed with an HFD for 16 weeks. (B) The representative gross liver images of these experimental mice. (C) LW, (D) LW/BW, and (E) ALT were determined in these mice. (F) The representative H&E images of the liver samples derived from these experimental mice. Left panel: Scale bars indicate 100 μm. Right panel: Scale bars indicate 50 μm. (G) Serum TG, (H) serum cholesterols, (I) hepatic cholesterols, and (J) hepatic TG were examined in these mice. Relative hepatic mRNA expression levels of (K) CD36, IGFBP2, (L) TET1, TET2, and TET3 were determined in these mice. Data are presented as mean ± S.D. in (A, C, D, E, G–K), and (L). P values were determined using unpaired two-tailed Student's T test, $n = 5$ in TET1[L/L]; $n = 15$ in Alb[Cre/-]TET1[L/L] for (A, C, D, E, G–K), and (L). Exact p values were as indicated. Source data are available online for this figure.

et al, 2021). In contrast, a study found that whole body TET1 KO promoted MASLD progression (Wang et al, 2020). Intriguingly, we used two whole body TET1 KO strategies and demonstrated that TET1 KO protects mice from MASLD progression. The possible explanation for this discrepancy is likely attributed to the transgenerational epigenetic inheritance given that TET1 KO would result in DNA methylation epigenetic changes (Fitz-James and Cavalli, 2022). Nevertheless, the impacts of liver TET1 on MASLD progression remain unclear. To fill the knowledge gap, our study revealed that liver specific TET1 KO suppressed MASLD progression

We systemically examined the DEGs in WT and TET1 KO liver samples and revealed that the most significant altered genes are related to lipid metabolic, fatty acid metabolic, and cellular lipid metabolic processes. Indeed, we found that hepatic and serum cholesterols were reduced in TET1 KO and TET1-LKO HFD mice.

Hepatic triglycerides decreased in TET1 KO HFD mice as well. Interestingly, we observed that visceral FW and FW/BW are increased in TET1 KO mice, contrasting the adipocyte TET1 KO mouse data (Damal Villivalam et al, 2020). However, visceral FW and FW/BW were not increased in TET1-LKO HFD mice compared to the control ones. The possible explanation for this discrepancy could be that one of the downregulated DEGs in TET1 KO HFD mice is peroxisome proliferator activated receptor gamma (PPARγ). Liver specific PPARγ deletion improved fatty liver but increased fat weights in leptin deficient mice(Matsusue et al, 2003). High energy expenditure and thermogenesis in TET1 KO mice might not fully explain these inconsistent findings since visceral FW and FW/BW are increased in these mice. Another potential explanation is that TET1 is required for the intestinal stem cell self-renewal and differentiation (Kim et al, 2016), both of which are important in mediating lipid homeostasis (Ko et al, 2020). Thus,

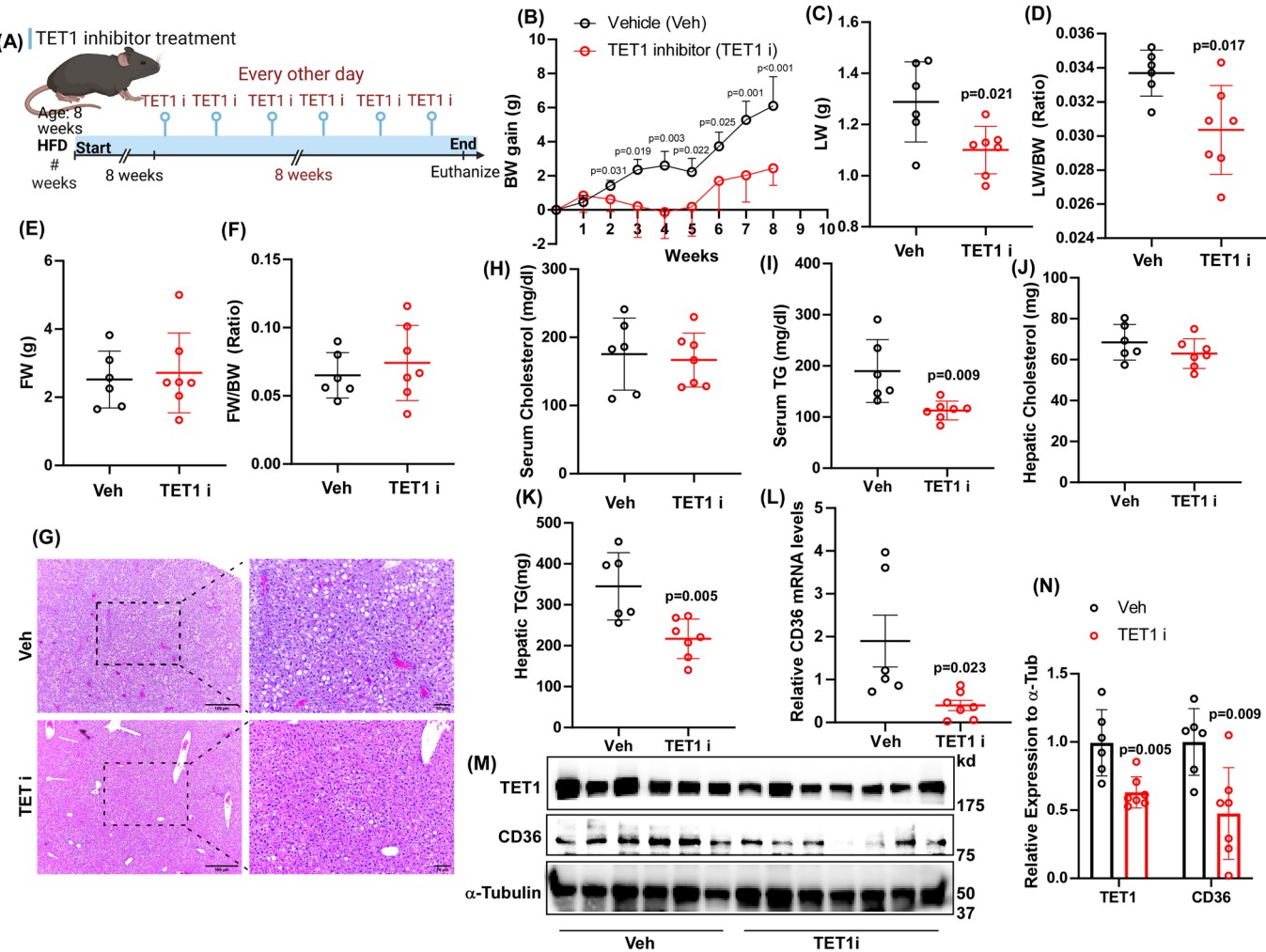

**Figure 7. Targeting TET1 with a small molecule inhibitor ameliorated MASLD progression.**

(A) The experimental scheme of TET1 inhibitor (TET1 i) treatment. (B) BW gains were measured in male C57BL/6J mice fed with an HFD and treated with a TET1 i (2.5 mg/kg UC514321) for 8 weeks. (C) LW, (D) LW/BW, (E) FW, (F) and FW/BW were determined in these mice. (G) The representative liver gross images of these experimental mice. Left panel: Scale bars indicate 100 μm, right panel: Scale bars indicate 50 μm. (H) Serum cholesterols, (I) serum TG, (J) hepatic cholesterol, and (K) hepatic TG in these mice. (L) The CD36 mRNA levels were examined in these mice. (M) TET1, CD36, and a-tubulin protein levels were examined in these mice. (N) Semi-quantification of TET1 and CD36 protein expression. Data are presented as mean ± S.D. in (B–F, H–L), and (N). P values were determined using unpaired two-tailed Student's T test, n = 6 in Veh; n = 7 in TET1 i for (B–F, H–L), and (N). Exact p values were as indicated. Source data are available online for this figure.

TET1 KO mice have impaired intestinal function which results in abnormal lipid homeostasis, leading to lipid storage in adipocytes. Given the complexity of organ cross talk in lipid metabolism, the role of TET1 in intestinal epithelial cell in MASLD warrant further investigation.

It has been described that CD36 hypomethylation is one of the reasons to cause MASLD progression in adult mice subjected to early life stress challenge (Fu et al, 2023). Besides, DNA hypomethylation of CD36 in primary human hepatocytes have been linked to the valproic acid-induced hepatic steatosis (van Breda et al, 2018). Thus, inducing CD36 DNA hypermethylation may lead to its downregulation, resulting in the improvement of MASLD. Interestingly, TET1 KO and TET1 LKO mice have reduced CD36 expression, indicating the reduced DNA

demethylation function of TET1 deficiency likely downregulating CD36 transcriptional activation. Given the consensus that CD36 upregulation is one of the major MASLD drivers (Rada et al, 2020), it would be important to evaluate whether inhibiting TET1 can ameliorate MASLD. Indeed, our in vivo studies demonstrated for the first time that targeting TET1 with a TET1 inhibitor significantly improved MASLD progression in mice.

In conclusion, using two whole body TET1 KO strategies, liver specific TET1 KO mice, and TET1 inhibitor treatment, we determined that suppressing TET1 ameliorated MASLD progression by inhibiting lipid metabolism through reducing fatty acid uptake. Our study clarifies the knowledge gap regarding how the hepatic TET1 is involved in MASLD progression and highlights the potential of targeting TET1 in hepatocytes to alleviate MASLD.

# Methods

### Reagents and tools table

| Reagent/Resource | Reference or Source | Identifier or Catalog Number |
| --- | --- | --- |
| **Experimental models** | | |
| C57BL/6J | Jackson lab | Strain #:000664 |
| B6;129S4-Tet1tm1.1Jae/J | Jackson lab | Strain #:017358 |
| B6.Cg-Speer6-ps1Tg(Alb-cre)21Mgn/J | Jackson lab | Strain #:003574 |
| B6N.FVB-Tmem163Tg(ACTB-cre)2Mrt/CjDswJ | Jackson lab | Strain #:019099 |
| TET1 floxed mice | PMID: 26199412, Dr. Anjana Rao | Not applicable |
| OUMS29 | PMID: 11513080 | Not applicable |
| HEK-293T | ATCC | CRL-3216™ |
| **Recombinant DNA** | | |
| shRNA-luciferase | Addgene | #78162 |
| shRNA-TET1#23 | Horizon | RHS3979-201792087 |
| shRNA-TER1#26 | Horizon | RHS3979-201797038 |
| pcDNA3.1-Flag | Addgene | #208051 |
| pcDNA3-Tet1 | Addgene | #60938 |
| p1D4-hrGFP II | Addgene | #71317 |
| pIRES-hrGFP II-mTET1 | Addgene | #83569 |
| pIRES-hrGFP II-TET1_CD | Addgene | #83570 |
| pIRES-hrGFP II-TET1 | Addgene | #83568 |
| pLenti-III-HA | Applied Biological Materials Inc. | LV587 |
| pLenti-III-HA-CD36 | Applied Biological Materials Inc. | 155340610295 |
| pGL4.24-Cd36-enhancer2 | Addgene | #138574 |
| pGL4.24-Cd36-enhancer3 | Addgene | #138575 |
| pGL4.24-Cd36-enhancer4 | Addgene | #138576 |
| pGL4.24-Cd36-enhancer5 | Addgene | #138577 |
| pGL4.24-Cd36-enhancer7 | Addgene | #138579 |
| pGL4.11-Cd36-promoter | Addgene | #138572 |
| **Antibodies** | | |
| TET1 Polyclonal antibody for human (1:1000) | ThermoFisherScientific | PA5-85489 |
| TET1 antibody for mouse (1:1000) | SigmaMillipore | SAB270073 |
| TET2 antibody (1:1000) | Active Motif | 61389 |
| TET3 antibody (1:1000) | Genetex | GTX121453 |
| GAPDH antibody (1:100,000) | ABclonal Technology | A19056 |
| Akt antibody (1:1000) | Cell Signaling Technology® | 4691S |
| pAkt antibody (1:1000) | Cell Signaling Technology® | 4060S |
| AMPK antibody (1:1000) | Cell Signaling Technology® | 2532S |
| pAMPK antibody (1:1000) | Cell Signaling Technology® | 2535S |
| PGC1a antibody (1:1000) | Cell Signaling Technology® | 2178 |
| a-tublin antibody (1:6000) | ABclonal Technology | A6830 |
| CD36 antibody (1:500) | Proteintech | 32181-1-AP |

| Reagent/Resource | Reference or Source | Identifier or Catalog Number |
| --- | --- | --- |
| LKB1 antibody (1:1000) | Cell Signaling Technology® | 3050S |
| pLKB1 antibody (1:1000) | Cell Signaling Technology® | 3482S |
| 5-methylcytosine antibody (1:1000) | Active Motif | 61479 |
| 5-hydroxylmethylcytosine antibody (1:10,000) | Active Motif | 39769 |
| **Oligonucleotides and other sequence-based reagents** | | |
| Human TET1-Forward (F) | Integrated DNA Technologies | CAGAACCTAAACCACCCGTG |
| Human TET1-Reverse (R) | Integrated DNA Technologies | TGCTTCGTAGCGCCATTGTAA |
| Human IGFBP2-F | Integrated DNA Technologies | CAGACGCTACGCTGCTATCC |
| Human IGFBP2-R | Integrated DNA Technologies | CCCTCAGAGTGGTCGTCATCA |
| Human CD36-F | Integrated DNA Technologies | GGCTGTGACCGGAACTGTG |
| Human CD36-R | Integrated DNA Technologies | AGGTCTCCAACTGGCATTAGAA |
| Mouse Tet1-F | Integrated DNA Technologies | ACACAGTGGTGCTAATGCAG |
| Mouse Tet1-R | Integrated DNA Technologies | AGCATGAACGGGAGAATCGG |
| Mouse Tet2-F | Integrated DNA Technologies | AGAGAAGACAATCGAGAAGTCGG |
| Mouse Tet2-R | Integrated DNA Technologies | CCTTCCGTACTCCCAAACTCAT |
| Mouse Tet3-F | Integrated DNA Technologies | TGCGATTGTGTCGAACAAATAGT |
| Mouse Tet3-R | Integrated DNA Technologies | TCCATACCGATCCTCCATGAG |
| Mouse Cd36-F | Integrated DNA Technologies | CCA GTC GGA GAC ATG CTT ATT |
| Mouse Cd36-R | Integrated DNA Technologies | GTA CAC AGT GGT GCC TGT T |
| Mouse Igfbp2-F | Integrated DNA Technologies | TGGAACATCTCTACTCCCTGC |
| Mouse Igfbp2-R | Integrated DNA Technologies | CTTCCCGGTATTGGGGTTCA |
| Mouse IGFBP3-F | Integrated DNA Technologies | CCA GGA AAC ATC AGT GAG TCC |
| Mouse IGFBP3-R | Integrated DNA Technologies | GGA TGG AAC TTG AAA TCG GTC A |
| Mouse Enho-F | Integrated DNA Technologies | CTC ATC GCC ATC GTC TGC AAT |
| Mouse Enho-R | Integrated DNA Technologies | GGG ACT GGA TTC CGA GAG AGA |
| Mouse Eda-F | Integrated DNA Technologies | CAA GGG TCA GCA ATT CAA GTC A |
| Mouse Eda-R | Integrated DNA Technologies | CAC CTT AGG GTT CAT AGT GAT GC |
| Mouse HSP90AA1-F | Integrated DNA Technologies | TGT TGC GGT ACT ACA CAT CTG C |
| Mouse HSP90AA1-R | Integrated DNA Technologies | GTC CTT GGT CTC ACC TGT GAT A |
| Mouse SLC27A1-F | Integrated DNA Technologies | TCT GTT CTG ATT CGT GTT CGG |
| Mouse SLC27A1-R | Integrated DNA Technologies | CAG CAT ATA CCA CTA CTG CGG |
| Mouse HNF4-A-F | Integrated DNA Technologies | GTG GCG AGT CCT TAT GAC ACG |
| Mouse HNF4-A-R | Integrated DNA Technologies | GCT GTT GGA TGA ATT GAG GTT GG |
| Mouse PPAR-A-F | Integrated DNA Technologies | TTT CGG CGA ACT ATT CGG CTG |

| Reagent/Resource | Reference or Source | Identifier or Catalog Number |
|---|---|---|
| Mouse PPAR-A-R | Integrated DNA Technologies | GGC ATT TGT TCC GGT TCT TCT T |
| Mouse CPT1b-F | Integrated DNA Technologies | TCT TCT TCC GAC AAA CCC TGA |
| Mouse CPT1b-R | Integrated DNA Technologies | GAG ACG GAC ACA GAT AGC CC |
| Mouse CPT1a-F | Integrated DNA Technologies | CTA TGC GCT ACT CGC TGA AGG |
| Mouse CPT1a-R | Integrated DNA Technologies | GGC TTT CGA CCC GAG AAG A |
| Mouse CrAT-F | Integrated DNA Technologies | CTC CTG GGC TGG AGT AGA TG |
| Mouse CrAT-R | Integrated DNA Technologies | TTA CAG AAG GGA CTG GAG CG |
| Mouse OCTN2-F | Integrated DNA Technologies | AAG ACC TGC AGG AAG CTG AA |
| Mouse OCTN2-R | Integrated DNA Technologies | TCC TTG TTT TTC GTG GGT GT |
| Mouse CYP4A10-F | Integrated DNA Technologies | TCC AGC AGT TCC CAT CAC CT |
| Mouse CYP4A10-R | Integrated DNA Technologies | TTG CTT CCC CAG AAC CAT CT |
| Mouse CYP4A14-F | Integrated DNA Technologies | TCA GTC TAT TTC TGG TGC TGT TC |
| Mouse CYP4A14-R | Integrated DNA Technologies | GAG CTC CTT GTC CTT CAG ATG GT |
| Mouse EHHADH-F | Integrated DNA Technologies | CGT CTC CTC GGT TGG TGT TC |
| Mouse EHHADH-R | Integrated DNA Technologies | ATT ATC TTC TTT GCA GTA TCT AGC TGC TT |
| Mouse FGF21-F | Integrated DNA Technologies | AAA GCC TCT AGG TTT CTT TGC CA |
| Mouse FGF21-R | Integrated DNA Technologies | CCT CAG GAT CAA AGT GAG GCG |
| **Chemicals, Enzymes and other reagents** | | |
| Mouse Diet, High Fat Fat Calories (60%), Soft Pellets | Bio-Serv | S3282 Pellets |
| Thermo Scientific Chemicals Methylene Blue, pure, certified | ThermoFisherScientific | A42771AP |
| Oil Red O | ThermoFisherScientific | AAA1298914 |
| Aspartate Aminotransferase Reagent Kit | ThermoFisherScientific | 23-666-121 |
| Alanine Aminotransferase Reagent | ThermoFisherScientific | 23-666-089 |
| Pointe Scientific Cholesterol Liquid Reagents | ThermoFisherScientific | 23-666-200 |
| Cholesterol Standard | ThermoFisherScientific | SB-1012-030 |
| Pointe Scientific Triglycerides Liquid Reagents | ThermoFisherScientific | 23-666-410 |
| BODIPY™ 581/591 C11 (Lipid Peroxidation Sensor) | ThermoFisherScientific | D3861 |
| Oleic acid | MilliporeSigma | O1383-5G |
| Dimethyl 2-oxoglutarate | MilliporeSigma | 13192-04-6 |
| Pointe Scientific Glucose (Hexokinase) Liquid Reagents | ThermoFisherScientific | 23-666-278 |
| UC-514321 | MedChemExpress | HY-120395 |
| Mouse Insulin ELISA Kit | ThermoFisherScientific | Invitrogen™ EMINS |
| **Software** | | |

| Reagent/Resource | Reference or Source | Identifier or Catalog Number |
|---|---|---|
| R-studio | Posit | RStudio 2024.04.1 + 748 |
| GraphPad Prism | Dotmatics | Version 10.4.1 |

## Animals

All animal studies were performed using male mice. It is well documented that male mice will develop more severe MASLD phenotypes than females. It is unknown whether the findings are relevant for female mice. The mice were housed in the animal facility of Tulane University School of Medicine, maintained under a 12-h light-dark cycle at 22 °C to 25 °C, with free access to standard laboratory mouse food and water. The male TET1 heterozygous KO (B6;129S4-Tet1tm1.1Jae/J) (Dawlaty et al, 2011), albumin Cre (B6.Cg-Speer6-ps1Tg(Alb-cre)21Mgn/J), and β-actin Cre (B6N.FVB-Tmem163Tg(ACTB-cre)2Mrt/CjDswJ) mice were purchased from the Jackson laboratory. The genotyping protocols for these mice were as the Jackson laboratory described, except TET1 KO. The forward and reverse primers used for TET1 wildtype (WT) and KO were GCTTCTCAGACTAGTGCTCTTC and AGAACCATCCAACTCACACTC, respectively. The PCR product sizes of TET1 WT and KO were 592 and 242, respectively. TET1 floxed mice (TET1$^{L/L}$) were kindly provided by Dr. Anjana Rao. The PCR genotyping protocols for floxed and deleted TET1 were performed as previously described (Kang et al, 2015). The floxed TET1, albumin Cre (B6.Cg-Speer6-ps1Tg(Alb-cre)21Mgn/J), and β-actin Cre (B6N.FVB-Tmem163Tg(ACTB-cre)2Mrt/CjDswJ) were generated with C57BL/6J background or backcrossed to C57BL/6J background more than 7 generations according to the original instruction manual. A high fat diet (60% calorie from fat, F3282, Bio-Serv) was used to establish MASLD mouse model. Basically, 8-weeks male mice were fed with the high fat diet (HFD) or normal chow (NC) for 16 weeks before collecting for analysis. For the UC514321 treatment, 8-weeks B6 mice were fed with HFD for 16 weeks and received the dose of 25 µg/kg UC514321 injection every other day during the last 8-weeks of HFD feeding. The same scheme of vehicle control injection was used for the control group. The animal studies were approved by the institutional animal care and use committee at Rhode Island Hospital and Tulane University School of Medicine, respectively. The ARRIVE guidelines were followed for this study.

## 5-hydroxymethylcytosine (5hmC) and 5-methylcytosine (5mC) dot blot assays

Genomic DNA of treated cells was prepared using phenol/chloroform extraction. Basically, cells were lysed with 500 µl DNA lysis buffer (10 mM Tris pH 8.0, 100 mM NaCl, 10 mM EDTA pH 8.0, 0.5% SDS in dH2O) plus 10 µl proteinase K (20 mg/ml) at 55 °C water bath overnight. The lysed cells were mixed with 200 µl solution of phenol:chloroform:isoamyl alcohol (25:24:1 saturated with 10 mM Tris, pH 8.0, 1 mM EDTA), shaked by hands for 3 min, and centrifuged at 14000 rpm for 3 min. 300 µl of supernatant were carefully transferred to a new tube, mixed with 300 µl isopropanol, incubated at room temperature for 20 min, and centrifuged at 12,000 rpm for 10 min. Supernatants were gently

discarded without losing DNA. DNA pellets were washed with 1 ml 75% ethanol and centrifuged at 12,000 rpm for 3 min. Discard supernatants and air dry the DNA for 20 min. DNA was dissolved in TE buffer and quantified accordingly. To do 5hmC dot blot assay, DNA was denatured at 100 °C for 10 min and rapidly cooled down on ice before applying to a GeneScreen Plus Hybridization Transfer Membrane (PerkinElmer). 1, 0.5, 0.25 µg of denatured DNA was blotted on a spot of the membrane, air dried for 30 min, and cross-linked with UV light. The membrane was blocked with 5% milk prepared in TBST at room temperature for 1 h and incubated with the 5hmC antibody (ActiveMotif) prepared in TBST (1:10,000 dilution) at 4 °C overnight. The membrane was washed with TBST for 5 min 3 times and then incubated with goat anti-rabbit antibody conjugated with HRP (1:10,000 dilution) for 1 h. The membrane was washed with TBST 10 min for 5 times and incubated with SuperSignal™ West Dura Extended Duration Substrate (Thermo Fisher Scientific). The images were taken using a Bio-Rad gel image system. 5hmC quantification was done using the image J software (NIH). For 5mC dot blot assay, it was performed using the same method except that the 5mC antibody (ActiveMotif) was used for overnight incubation.

For methylene blue staining, 1, 0.5, 0.25 µg of non-denatured DNA was blotted on a new membrane, air dried for 30 min, and fixed with the UV light. The membrane was stained with methylene staining solution (0.1% methylene in 0.5 M sodium acetate, pH 5.2) for 15 min. The membrane was washed in deionized $H_2O$ for 3 min 5 times. Images were taken and quantified as 5hmC ones.

## Free fatty acid uptake

The $1 \times 10^5$ OUMS29 cells treated as described accordingly were sub-cultured in 60 mm cell culture dish with 6 ml complete culture medium. One dish was labeled as the control group without staining. 24 h later, 6 µL of 1 mM BODIPY-C12 stock solution were added to the culture dish, mixed well by shaking/rocking the plates, and incubated with cells for 20 min in the cell culture incubator. After incubation, the cell culture medium containing suspended cells were transferred to a 15 mL tubes. The attached cells were washed with 2 mL PBS, but the PBS supernatants were collected in the same 15 mL tubes. Then, cells were trypsinzed using 2 ml trypsin. 2 ml cell culture medium were used to neutralize trypsin. All detached cells were collected in the same 15 mL tubes and centrifuged at $220 \times g$ for 5 min. Supernatants were discarded and cell pellets were washed using 2 mL PBS for 3 times. After that, cells were resuspended with 500 µL PBS. Cells were transferred to a 5 ml polystyrene round bottom tube a with a cell strainer cap and centrifuged at $220 \times g$ for 4 min to remove cell aggregates. Cells were subjected to the flow cytometry analysis according to the BODIPY-C12 instruction manual (D3822, ThermoFisher Scientific, Waltham, MA). The collected flow cytometry data were then analyzed using the FlowJo software (Ashland, OR).

## RNA sequencing (RNA-Seq)

The mRNA sequencing was performed using the ArraryStar service. 10 micrograms of mRNA samples were collected and shipped to the ArraryStar for sequencing experiments.

RNA-seq data were loaded using the readxl package in R studio which included gene expression levels, log2 fold change

(log2FoldChange), and adjusted $p$-values (p-adj). Differentially expressed genes (DEGs) were identified using thresholds of p-adj < 0.05 and log2FoldChange between −3 and 3 based on biological relevance and statistical consideration and then top 50 DEGs were selected. DEGs were categorized as upregulated or downregulated in TET1 KO mice compared to the WT ones based on the significance and their log2FoldChange values. Genes with log2FoldChange > 0 were labeled as "Upregulated in TET1 KO," while those with log2FoldChange < 0 were labeled as "Down-regulated in TET1 KO". A heatmap was generated to visualize the expression patterns of the top 50 DEGs across samples. Count data for the selected genes were extracted and converted into a matrix. Samples were reordered so that WT samples appeared on the left and TET1 KO samples on the right. The heatmap was generated using the pheatmap package, with rows (genes) scaled and hierarchical clustering applied to both rows and columns. Sample annotations were added to distinguish between WT and TET1 KO conditions. A volcano plot was created to visualize the distribution of all genes, highlighting the top 50 DEGs. The x-axis represented log2FoldChange (TET1 KO vs WT), and the y-axis represented −log10(p-adj). All genes were plotted as gray points, while the top 500 was highlighted as orange and the top 50 DEGs regulated genes were highlighted in red. Gene symbols for the top 10 DEGs (upregulated and downregulated each) were labeled on the plot. Gene Ontology (GO) enrichment analysis was performed separately for upregulated and downregulated genes using the clusterProfiler package. Gene symbols were converted to Entrez IDs using the org.Mm.eg.db database. Enriched GO terms were identified for the Biological Process (BP) ontology, with significant thresholds of indicated $p$-values. Redundant GO terms were simplified to improve interpretability.

## Lipidomic data analysis

The lipidomic analysis was performed via LIPEA (Lipid Pathway Enrichment Analysis) and KEGG (Kyoto Encyclopedia of Genes and Genomes) pathway analysis tools. Processed lipid data, including lipid species and their corresponding fold changes, was fed into the LIPEA tool. The top enriched pathways were identified based on statistical significance ($p$-value < 0.05), and the highly enriched pathways were further filtered and validated using the KEGG pathway database. This approach allowed for the identification of key lipid metabolic pathways, which were significantly altered in the TET1 LKO and TET1 KO groups compared to WT.

## ChIP analysis

The computational analysis was conducted within the R-studio environment (R version 4.4.0), integrating GEO datasets "GSE26832 (Wu et al, 2011)", "GSE140999 (Huang et al, 2020)", and "GSM611192 (Williams et al, 2011)". The base script provided by Bioconductor was followed with necessary modifications. Specifically, the chipseq library facilitated ChIP-seq analysis by tiling the mouse genome (assembly mm9) into bins ranging from 200 bp to 400 bp, utilizing the strand information object (si). The regulatory architecture of the CD36 gene was delineated by annotating promoter and enhancer regions using the Gviz library. Additionally, the Gviz and biomaRt libraries were employed to annotate and align with the CD36 gene, through connection to the Ensembl database. The results were presented by plotting the histogram.

## Luciferase assay

Luciferase analysis was performed as previously described (Huang et al, 2013). The luciferase plasmids of pGL4.11-Cd36-promoter, pGL4.24-Cd36-enhancer1, 2, 3, 4, 5, 6, 7, and 8 were gifts from Lora Hooper (Kuang et al, 2019) (Addgene plasmid # 138572, 138573, 138574, 138575, 138576, 138577, 138578, 138579, and 138580). pcDNA3-Tet1 was a gift from Yi Zhang (Wang and Zhang, 2014) (Addgene plasmid # 60938). pIRES-hrGFP II-TET1 was a gift from Jean-Pierre Issa (Jin et al, 2014) (Addgene plasmid # 83568).

## Cell culture

Immortalized human hepatocytes, OUMS29 cells (Huang et al, 2015) were grown in Dulbecco's modified Eagle medium with 10% fetal bovine serum (Gibco), 2 mM L-glutamine (Gibco), and Penicillin-Streptomycin (100 U/mL) (Lonza). TET1 knockdown OUMS29 cells were generated using a lentiviral transduction strategy as previously described (Bai et al, 2021). Cells were tested negative for mycoplasma contamination and authenticated.

## Immunoblot analysis

Immunoblot (IB) was performed as previously described (Huang et al, 2015). 50 micrograms of total protein lysates were used for all IB experiments. Specific antibodies were used to incubate with the target proteins blotted on the Polyvinylidene fluoride (PVDF) membrane at 4 °C overnight. The secondary antibodies, either anti-rabbit (PI-1000-1, Vector Lab) or anti-mouse (PI-2000-1, Vector Lab), conjugated with peroxidase were used to incubate with the PVDF membrane at room temperature for 1 h. Then, the membrane was washed with 1X Tris-Buffered Saline containing 0.05% Tween® 20 (TBST) for 5 min 3 times. After that, the membrane was incubated with the SuperSignal™ West Dura Extended Duration Substrate (#34076, ThermoScientific) and imaged using the ChemiDoc system (Bio-Rad).

## Liver functional assay

The Alanine aminotransferase (ALT) and aspartate aminotransferase (AST) were determined for liver functions by using the AST and ALT kits as previously described (Nagaoka et al, 2020).

## Cholesterol, triglyceride, and glucose analysis

Mouse serum samples were collected at the end of 16-week HFD feeding without fasting. The liver tissue lysates were prepared using the RIPA buffer without adding proteinase and phosphatase inhibitors. Cholesterol and triglyceride levels were determined in mouse serum samples and 50 microgram liver tissue lysates using the assay kits as previously described (Nagaoka et al, 2020). Glucose levels were determined in mouse serum samples using the Pointe Scientific Glucose (Hexokinase) Liquid Reagents (23-666-278).

## Histological assays

The paraffin-embedded liver tissue slides were deparaffined, rehydrated, and subjected to the hematoxylin and eosin (H&E) staining. For the Oil Red O staining, it was performed using frozen liver tissue sections and the Oil Red Stain kit (ab150678, Abcam).

### The paper explained

**Problem**

The metabolic dysfunction-associated steatotic liver disease can progress to liver fibrosis and further advance to cirrhosis and hepatocellular carcinoma, which are the 12th leading cause of death and the 5th leading cause of cancer-associated death. Understanding the molecular mechanisms underlying MASLD progression may help identify potential therapeutic targets.

**Results**

DNA methylation, a type of DNA modification carried by two families of proteins including TET1, generally suppresses mRNA expression. Global liver DNA methylation change has been linked to MASLD progression in humans. We found that TET1 is decreased in MASLD, and depleting TET1 protects MASLD progression in a preclinical model. Depleting TET1 in the liver resulted in improved MASLD as well. It was found that TET1 promoted the transcription of CD36 which helps free fatty acid uptake in the liver. Excessive free fatty acid uptake in the liver would lead to MASLD progression. Targeting TET1 in human hepatocytes with molecular and biochemical approaches both led to reduced free fatty acid uptake. More importantly, treating MASLD mice with a small molecule to inhibit TET1 function significantly alleviated MASLD progression.

**Impact**

Our study clarifies the knowledge gap regarding how hepatic TET1 is involved in MASLD progression and highlights the potential of targeting TET1 in hepatocytes to alleviate MASLD.

## Oil Red O staining

Cells were cultured to the desired confluence in Dulbecco's modified Eagle medium (DMEM). Prior to staining, the medium was aspirated, and the cells were washed with phosphate-buffered saline (PBS). The cells were then fixed with 10% formalin for 15 min at room temperature. After fixation, they were washed with 60% isopropanol and allowed to air dry. The Oil Red O working solution was prepared by mixing the stock solution with distilled water in a 3:2 ratio. The lipid droplets were stained with the working solution of Oil Red O for 30 min. Excess stain was removed by washing the cells three to four times with distilled water. The cells were then examined under an inverted microscope to assess the presence and distribution of lipid droplets.

## Statistical analysis

Results from two groups were analyzed using the student t test to determine statistical difference. If more than two groups were involved in the studies, the one-way ANOVA analysis was adopted for the statistical analysis. The experimental replicates are indicated in the graph and figure legend. In vivo and in vitro studies were performed in biological and technical repeats, respectively.

Sample sizes were determined based on preliminary data and power analysis to ensure adequate statistical power. Exclusion criteria were predefined to remove outliers or data points that did not meet quality control standards, while inclusion criteria ensured consistency across experimental groups. Randomization was applied to assign samples or animals to experimental groups to minimize bias. Blinding was implemented during data collection

and analysis to ensure objectivity. Statistical significance was set at $p < 0.05$.

### Graphics

Schematic diagrams and graphical representations included in this manuscript were created using BioRender.com.

## Data availability

The RNA-Seq dataset, GSE286394, is publicly available in GEO databases.

The source data of this paper are collected in the following database record: biostudies:S-SCDT-10_1038-S44321-025-00224-4.

## Peer review information

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

## Acknowledgements

Funding for this work was provided by 2017 AASLDF Pinnacle Research Development Award (C-KH), P20GM109035 pilot grant (C-KH), R01 DK135664 (C-KH), and R21AA028576 (C-KH), R01AA030424 (WF), and R01AA023190 (WF). We want to thank Dr. Joselynn Wallace, who is the Senior Genomics Data Scientist Center for Computation and Visualization at Brown University, for RAN-Seq raw data processing. We also want to thank Connie Porretta who is the director of Flow Cytometry Shared Resource Lab at Tulane University School of Medicine for Flow Cytometry experiments.

## Author contributions

**Hongze Chen**: Data curation; Formal analysis; Validation; Investigation; Methodology; Writing—original draft. **Muhammad Azhar Nisar**: Conceptualization; Data curation; Software; Formal analysis; Validation; Investigation; Methodology; Writing—original draft. **Joud Mulla**: Data curation; Formal analysis; Validation; Investigation; Methodology. **Xinjian Li**: Data curation; Formal analysis; Validation; Investigation. **Kevin Cao**: Resources; Data curation; Investigation; Methodology. **Shaolei Lu**: Conceptualization; Resources; Funding acquisition; Methodology; Writing—original draft; Writing—review and editing. **Katsuya Nagaoka**: Data curation; Validation; Investigation; Visualization; Methodology. **Shang Wu**: Data curation; Formal analysis; Validation; Investigation; Methodology. **Peng-Sheng Ting**: Conceptualization; Resources; Data curation; Methodology; Writing—original draft. **Tung-Sung Tseng**: Resources; Software; Formal analysis; Investigation; Methodology. **Hui-Yi Lin**: Resources; Formal analysis; Investigation; Methodology. **Xiao-Ming Yin**: Resources; Formal analysis; Funding acquisition; Writing—original draft. **Wenke Feng**: Resources; Funding acquisition; Investigation; Writing—original draft. **Zhijin Wu**: Software; Funding acquisition; Methodology. **Zhixiang Cheng**: Data curation; Investigation; Methodology. **William Mueller**: Data curation; Formal analysis; Investigation; Methodology. **Amalia Bay**: Data curation; Investigation; Methodology. **Layla Schechner**: Data curation; Investigation; Methodology. **Xuewei Bai**: Data curation; Investigation; Methodology. **Chiung-Kuei Huang**: Conceptualization; Resources; Data curation; Formal analysis; Supervision; Funding acquisition; Investigation; Methodology; Writing—original draft; Project administration; Writing—review and editing.

Source data underlying figure panels in this paper may have individual authorship assigned. Where available, figure panel/source data authorship is listed in the following database record: biostudies:S-SCDT-10_1038-S44321-025-00224-4.

## Disclosure and competing interests statement

Dr. Xiao-Ming Yin is the Board of Advisor, NanoPin Technologies, New Orleans, LA. The NanoPin Technologies has no role in this study. All other authors have no competing interests.

