## [Peer Review File · EMBO Molecular Medicine]

Liver TET1 promotes metabolic dysfunction-associated steatotic liver disease

Hongze Chen, Muhammad Nisar, Joud Mulla, Xinjian Li, Kevin Cao, Shaolei Lu, Katsuya Nagaoka, Shang Wu, Peng-Sheng Ting, Tung-Sung Tseng, Hui-Yi Lin, Xiao-Ming Yin, Wenke Feng, Zhijin Wu, Zhixiang Cheng, William Mueller, Amalia Bay, Layla Schechner, Xuewei Bai, and Chung-Kuei Huang

Corresponding author(s): Chung-Kuei Huang (chuang17@tulane.edu)

Review Timeline:

Submission Date:	12th Sep 24
Editorial Decision:	16th Oct 24
Revision Received:	17th Jan 25
Editorial Decision:	7th Feb 25
Revision Received:	7th Mar 25
Accepted:	12th Mar 25

Editor: Lise Roth

Transaction Report:

16th Oct 2024

Dear Dr. Huang,

Thank you for the submission of your manuscript to EMBO Molecular Medicine, and please accept my apologies for the delay in getting back to you as we were waiting for one additional referee report. Unfortunately, referee #1 has not gotten back to us despite several chasers, but given that both referees #2 and #3 provide similar recommendations, we prefer to make a decision now in order to avoid further delay in the process. Should referee #1 provide a report, we will send it to you, with the understanding that we will not ask you extensive experiments in addition to the ones required in the enclosed reports from referees #2 and #3.

As you will see from the reports below, both referees mention the novelty and potential translational interest of the work, however they also highlight several concerns that should be addressed, including (but not limited to) effect of TET1 deletion on body weight gain, on diabetes and obesity, and unclear mechanism.

Addressing these points and the other reviewers' concerns in full will be necessary for further considering the manuscript in our journal, and acceptance of the manuscript will entail a second round of review. EMBO Molecular Medicine encourages a single round of revision only and therefore, acceptance or rejection of the manuscript will depend on the completeness of your responses included in the next, final version of the manuscript. For this reason, and to save you from any frustrations in the end, I would strongly advise against returning an incomplete revision.

We are expecting your revised manuscript within three months, if you anticipate any delay, please contact us.

We require:

4) A .docx formatted letter INCLUDING the reviewers' reports and your detailed point-by-point responses to their comments. As part of the EMBO Press transparent editorial process, the point-by-point response is part of the Review Process File (RPF), which will be published alongside your paper.

5) A complete author checklist, which you can download from our author guidelines (<https://www.embopress.org/page/journal/17574684/authorguide#submissionofrevisions>). Please insert information in the checklist that is also reflected in the manuscript. The completed author checklist will also be part of the RPF.

6) All Materials and Methods need to be described in the main text using our 'Structured Methods' format. According to this format, the Methods section includes a Reagents and Tools Table (listing key reagents, experimental models, software and relevant equipment and including their sources and relevant identifiers) followed by a Methods and Protocols section describing the methods, ideally using a step-by-step protocol format. The aim is to facilitate adoption of the methodologies across labs. Please download and fill our Reagents and Tools Table template (.docx), which you can find in our author guidelines: <https://www.embopress.org/page/journal/14693178/authorguide#structuredmethods>.

<https://www.embopress.org/doi/10.15252/msb.20178071>

7) Please note that all corresponding authors are required to supply an ORCID ID for their name upon submission of a revised manuscript.

8) It is mandatory to include a 'Data Availability' section after the Materials and Methods. Before submitting your revision, primary datasets produced in this study need to be deposited in an appropriate public database, and the accession numbers and database listed under 'Data Availability'. Please remember to provide a reviewer password if the datasets are not yet public (see <https://www.embopress.org/page/journal/17574684/authorguide#dataavailability>).

9) For data quantification: please specify the name of the statistical test used to generate error bars and P values, the number (n) of independent experiments (specify technical or biological replicates) underlying each data point and the test used to calculate p-values in each figure legend. The figure legends should contain a basic description of n, P and the test applied. Graphs must include a description of the bars and the error bars (s.d., s.e.m.). Please provide exact p values.

10) Our journal encourages inclusion of *data citations in the reference list* to directly cite datasets that were re-used and obtained from public databases. Data citations in the article text are distinct from normal bibliographical citations and should directly link to the database records from which the data can be accessed. In the main text, data citations are formatted as follows: "Data ref: Smith et al, 2001" or "Data ref: NCBI Sequence Read Archive PRJNA342805, 2017". In the Reference list, data citations must be labeled with "[DATASET]". A data reference must provide the database name, accession number/identifiers and a resolvable link to the landing page from which the data can be accessed at the end of the reference. Further instructions are available at .

11) We replaced Supplementary Information with Expanded View (EV) Figures and Tables that are collapsible/expandable online. A maximum of 5 EV Figures can be typeset. EV Figures should be cited as 'Figure EV1, Figure EV2' etc... in the text and their respective legends should be included in the main text after the legends of regular figures.

12) The paper explained: EMBO Molecular Medicine articles are accompanied by a summary of the articles to emphasize the major findings in the paper and their medical implications for the non-specialist reader. Please provide a draft summary of your article highlighting

- the medical issue you are addressing,

- the results obtained and

- their clinical impact.

13) Author contributions: CRedit has replaced the traditional author contributions section because it offers a systematic machine readable author contributions format that allows for more effective research assessment. Please remove the Authors Contributions from the manuscript and use the free text boxes beneath each contributing author's name in our system to add specific details on the author's contribution. More information is available in our guide to authors.

Please also suggest a visual abstract to illustrate your article as a PNG file 550 px wide x 300-600 px high. A cropped portion of this image will serve as thumbnail for the table of content on our webpage.

16) As part of the EMBO Publications transparent editorial process initiative (see our Editorial at <http://embomolmed.embopress.org/content/2/9/329>), EMBO Molecular Medicine will publish online a Review Process File (RPF)

to accompany accepted manuscripts.

In the event of acceptance, this file will be published in conjunction with your paper and will include the anonymous referee reports, your point-by-point response and all pertinent correspondence relating to the manuscript. Let us know whether you agree with the publication of the RPF and as here, if you want to remove or not any figures from it prior to publication. Please note that the Authors checklist will be published at the end of the RPF.

I look forward to receiving your revised manuscript.

Yours sincerely,

Lise Roth

***** Reviewer's comments *****

Referee #2 (Remarks for Author):

This manuscript details the effects of Liver TET1 on the progression of MASLD using full body and liver-specific TET1 KO, in vitro studies and an analysis of published chip-seq databases. Overall, the findings are novel, the analysis are well thought out and the data are presented in a clear way.

I have the following comments/suggestions:

- What is the reason for the lower body weight gain in liver/whole body TET1 knock-out mice? I assume lower uptake of fatty acids in the liver does not lead to generalised weight loss. Is there a higher energy expenditure - can the mice be analysed by in metabolic cages?
- From the CHIP-seq experiments described, it is unclear to me if TET1 upregulated CD36 through direct binding with the enhancers, or if the mechanism is purely linked to demethylation of methylated cytosines in the enhancer regions. Can this be mechanistically shown? If the former, what is the link with the function of TET1 as an unmethylater?
- The metabolic pathways affected by TET1 KO, related to lipid metabolism, are vague. A lipidomic analysis of the liver could contribute to a better understanding of specific changes in hepatic fat content.

Minor:

- Please indicate on the figures whether hepatic or serum cholesterol/triglycerides are displayed. I know this is detailed in the legends, but it would be helpful if the figures also show this.
- What was the genetic background (strain) of the KO mice? The mouse experimental circumstances can be described in more detail: housing, temperature, humidity, day-/night cycle... Please follow the ARRIVE guidelines.
- The authors write that 'the global methylation was found elevated in the HFD-treated mice. How does this compare to a global demethylation proposed in the introduction? Is this a topic of debate in the literature?
- The level of English is fine, but there are some typos or minor grammatical mistakes in the text.

Referee #3 (Remarks for Author):

This is a very interesting manuscript showing the effect of Tet1 on liver fatty acid homeostasis and its putative role in MASLD.

There are several aspects of the manuscript that can be improved:

- whole body metabolism is little explored while Tet1 seems to influence body weight gain. What is the impact of Tet1 on food

intake and adipose tissues. It would also be interesting to explore hepatokine expression and their potential correlations with changes in wholebody energy homeostasis and/or changes in food intake.

- Tet1 regulates glucose homeostasis (Zhang et al., *elife*, 2021). There is a strong correlation between diabetes and MASLD. Is the effect of liver Tet1 on MASLD associated with changes in glucose homeostasis? In other words, does Tet1 deletion in hepatocytes not only influence MASLD but also obesity and diabetes.

- Changes in gene expression should be investigated with more details. The figure 3 and its legend must be improved.

The number of replicates (it seems from the heatmap that there are only 3 samples analyzed for gene expression) is not adequate for a thorough bioinfo and biostat analysis.

Most regulated genes should be presented (perhaps a volcano plot).

Since the authors suggest that Tet1 mediated regulation of CD36 gene expression is involved in fatty acid uptake and MASLD, it would be great to specifically investigate the effect of Tet1 on transcription factors that act as fatty acid sensors. In my opinion, there are two of these: HNF4alpha (simcox et al., *Cell metab*, 2017) and PPARalpha (fougerat et al., *Cell rep*, 2022).

Referee #1

Comment#1. This manuscript details the effects of Liver TET1 on the progression of MASLD using full body and liver-specific TET1 KO, in vitro studies and an analysis of published chip-seq databases. Overall, the findings are novel, the analysis are well thought out and the data are presented in a clear way.

Response: Thank you for taking the time to review our manuscript and for your positive feedback. We appreciate your recognition of the novelty of our findings, the thoroughness of our analysis, and the clarity of our data presentation. Your comments have further helped us improve the manuscript.

Comment #2. What is the reason for the lower body weight gain in liver/whole body TET1 knock-out mice? I assume lower uptake of fatty acids in the liver does not lead to generalised weight loss. Is there a higher energy expenditure - can the mice be analysed by in metabolic cages?

Response: We completely agree with the reviewer that decreased fatty acid uptake in the liver specific TET1 knockout (TET1 LKO) mice should lead to improved hepatic steatosis but not reduced body weight gain. We measured food consumption in TET1 LKO, TET1 KO, and the littermate control mice fed with an HFD. However, we did not observe significant difference regarding food consumption (SFig. 4A). Thus, we measured fed blood glucose and insulin levels. We found that blood glucose is not different between control and TET1 KO and TET1 LKO mice (SFig. 4C, D, and SFig. 8D). However, insulin levels were decreased in TET1 KO and TET1 LKO mice (SFig. 4E and SFig. 8E), suggesting that improved glucose homeostasis in TET1 LKO and TET1 KO mice is consistent to reduced weight gain. We further analyzed all hepatokines related to body weight change in control and TET1 KO mice. We found that IGFBP2, IGFBP3, Enho, Eda, and HSP90aa were significantly altered in TET1 KO mice (Supplemental Table 2). We also examined these genes in control and TET1 LKO mice (SFig. 5). We only found that Enho is upregulated in TET1 LKO mice. Interestingly, it has been previously reported that transgenic overexpression of Enho improves diet induced obesity, glucose homeostasis, and fatty liver (PMID: 19041763). Based on these data, it is very likely that the reduced body weight gain in TET1 LKO and TET1 KO mice is highly related to elevated Enho expression.

Comment #3. From the CHIP-seq experiments described, it is unclear to me if TET1 upregulated CD36 through direct binding with the enhancers, or if the mechanism is purely linked to demethylation of methylated cytosines in the enhancer regions. Can this be mechanistically shown? If the former, what is the link with the function of TET1 as an unmethylater?

Response: Thanks for the reviewer's comment. It is essential to understand how CD36 is regulated by TET1. In this case, we performed CD36 enhancer luciferase experiments by co-transfecting enhancer-luciferase with empty vector control, pIRES-hrGFP II-TET1 (full-length

TET1), pIRES-hrGFP II-mTET1 (full-length TET1 with catalytic domain mutation), and pIRES-hrGFP II-TET1_CD (TET1 catalytic domain only). We found that the transfections of full-length TET1 and TET1 catalytic domain only promoted CD36 enhancer transactivation but not full-length TET1 with catalytic domain mutation (Fig. 5F). Besides, the treatment of DNA methylation inhibitor, 5-Azacytidine could significantly reverse the effects of TET1 knockdown on downregulating CD36 in human hepatocytes (Fig. 5G). All these results lead to the conclusion that TET1 upregulates CD36 mainly through promoting DNA demethylation.

Comment #4. The metabolic pathways affected by TET1 KO, related to lipid metabolism, are vague. A lipidomic analysis of the liver could contribute to a better understanding of specific changes in hepatic fat content.

Response: Thank you for your valuable feedback regarding the metabolic pathways affected by TET1 KO and their relation to lipid metabolism. We agree that a more detailed analysis would enhance the understanding of specific changes in hepatic fat content. To address this, we have conducted a lipidomic analysis of the liver. The lipidomic assay and enrichment analysis of lipidomic data, performed using Lipid Pathway Enrichment Analysis (LIPEA) and Kyoto Encyclopedia of Genes and Genomes (KEGG) tools, reveals that the most significant altered pathway in the TET1 LKO and TET1 KO groups compared to the control one is glycerophospholipid metabolism. The key upregulated lipids in this pathway are 1-Acyl-sn-glycero-3-phosphocholine, 2-Acyl-sn-glycero-3-phosphocholine, 1-Acyl-sn-glycero-3-phosphoethanolamine, and 2-Acyl-sn-glycero-3-phosphoethanolamine. This leads to the significant enrichment of both the phosphatidylethanolamine and phosphatidylcholine biosynthesis pathways. Conversely, the downregulated lipids in the glycerophospholipid metabolism pathway are 1-Acyl-sn-glycerol-3-phosphate and 2-Acyl-sn-glycerol-3-phosphate (Supplemental table 3-5 and SFig. 6A and B).

Comment #5. Please indicate on the figures whether hepatic or serum cholesterol/triglycerides are displayed. I know this is detailed in the legends, but it would be helpful if the figures also show this.

Response: Clear illustration of hepatic and serum cholesterol and triglyceride levels would help audiences understand the published article. Thus, we followed the reviewer's suggestion to revise our figures accordingly.

Comment #6. What was the genetic background (strain) of the KO mice? The mouse experimental circumstances can be described in more detail: housing, temperature, humidity, day-/night cycle... Please follow the ARRIVE guidelines.

Response: We have described all mouse strains in supplemental information. For all mice, they were established with C57BL/6J background. The housing environment of these experimental mice was also described in detail in supplemental information.

Comment #7. The authors write that 'the global methylation was found elevated in the HFD-treated mice. How does this compare to a global demethylation proposed in the introduction? Is this a topic of debate in the literature?

Response: It was previously demonstrated that global 5hmC is slightly decreased in mice fed with an HFD (PMID: 31389294), indicating that downregulated 5hmC should likely lead to elevated DNA methylation which we and other groups both observed in HFD fed mice (PMID: 37282749). However, clinical studies suggested that low DNA methylation is associated with MASLD progression in obese population (PMID: 31785086). Interestingly, another study suggested that the co-substrate of TETs, alpha-ketoglutarate, is positively associated with MASLD progression. Alpha-ketoglutarate levels increased 6-17 times in MASLD patients when compared with the lean ones (PMID: 24675715). Given that elevated TET enzymatic activity by high alpha-ketoglutarate levels would lead to DNA demethylation resulting in low DNA methylation, it is conceivable that low DNA methylation is associated with MASLD progression. Unexpectedly, we found that TET1 and TET3 protein levels were substantially decreased in HFD mice (Fig. 1F), suggesting that upregulated DNA methylation in mice is likely due to reduced TET1 and TET3 protein expression levels.

Comment #8. The level of English is fine, but there are some typos or minor grammatical mistakes in the text.

Response: Sorry for the issues. To address these concerns, we have asked two native speakers, who major in biomedical science, to proofread our manuscript.

Referee #2

Comment #1. This is a very interesting manuscript showing the effect of Tet1 on liver fatty acid homeostasis and its putative role in MASLD.

Response: Thank you so much for the reviewer's positive comment about our findings.

Comment #2. There are several aspects of the manuscript that can be improved: Whole body metabolism is little explored while Tet1 seems to influence body weight gain. What is the impact of Tet1 on food intake and adipose tissues. It would also be interesting to explore hepatokine expression and their potential correlations with changes in whole body energy homeostasis and/or changes in food intake.

Response: Thanks for reviewer's comment. While we focused on the liver TET1 role on MASLD progression, it is necessary to determine how TET1 affects whole body metabolism in order to clarify how liver TET1 promotes MASLD progression. We found that whole body TET1 KO increases visceral fat weight but not liver specific TET1 KO mice (Fig. 2D and SFig. 8B). Interestingly, both whole body and liver specific TET1 KO have no significant impacts on food consumption (SFig. 4A and SFig. 8A). When we analyzed hepatokines in control and TET1 KO

mice, we found that IGFBP2, IGFBP3, Enho, Eda, Slc27a1, and HSP90aa1 were significantly altered in TET1 KO mice (Supplemental Table 2). Among them, we found elevated IGFBP2 and Enho expression levels were correlated with improved MASLD in TET1 KO mice. We further analyzed IGFBP2 and Enho in TET1 LKO mice. However, we only observed the upregulation of Enho in TET1 LKO mice. Interestingly, we also observed improved glucose homeostasis in TET1 KO and TET1 LKO mice (SFig. 4E and SFig. 8E). Further, a previous study has demonstrated that glucose homeostasis and diet induced obesity were improved in Enho transgenic mice (PMID: 19041763), suggesting that the liver TET1-mediated obesity phenotype is highly related to Enho change.

Comment #3. Tet1 regulates glucose homeostasis (Zhang et al., *elife*, 2021). There is a strong correlation between diabetes and MASLD. Is the effect of liver Tet1 on MASLD associated with changes in glucose homeostasis? In other words, does Tet1 deletion in hepatocytes not only influence MASLD but also obesity and diabetes.

Response: Based on the reduced body weight data in TET1 LKO mice, we believe that liver TET1 is likely involved in glucose homeostasis since no food consumption difference was found between these mice (SFig. 8A). Besides, visceral fat weight is not changed in TET1 LKO mice when compared to the control ones (SFig. 8B and C). Interestingly, we found that fed insulin is slightly decreased in TET1 KO and TET1 LKO groups compared to the control ones (SFig. 4E and SFig. 8E). Besides, diet induced obesity is suppressed by TET1 KO and TET1 LKO as well. Therefore, liver TET1 deletion improved glucose homeostasis and diet induced obesity.

Comment #4. Changes in gene expression should be investigated with more details. The figure 3 and its legend must be improved.

Responses: Thanks for the reviewer's comment. To further investigate the details of gene expression changes, we generated a differentially expressed gene (DEG) table in the supplemental table 1. Besides, we generated a heat map listing the top 50 altered genes. We then investigated the DEGs using gene set enrichment analysis with the Gene Ontology database. We found that many lipid metabolism pathways were downregulated in TET1 KO mice (Fig. 3G-I). We also followed the reviewer's suggestions to elaborate the details of figure legend for figure 3.

Comment #5. The number of replicates (it seems from the heatmap that there are only 3 samples analyzed for gene expression) is not adequate for a thorough bioinfo and biostat analysis.

Response: We respectfully disagree with this reviewer's comment that the number of replicates is not adequate for a thorough bioinfo and biostat analysis. The purpose of our RNA sequencing (RNA-Seq) experiment is to clarify the potential downstream targets involved in MASLD progression mediated by TET1 knockout. We generated the RNA-Seq data using 3 wildtype and 4 TET1 KO mice since we only have 3 wildtype and 4 TET1 KO liver samples ready while preparing mRNA for sequencing. We were able to clarify that many lipid metabolic genes were substantially and significantly altered in TET1 KO mice. Among them, we determined that TET1

likely promoted MASLD expression through upregulating CD36. With a low mouse number, we might miss some potential TET1 downstream targets involved in MASLD progression. Nevertheless, the other potentially TET1-mediated downstream genes could be likely generated and concluded from high mouse numbers due to marginal change or huge variation. In this case, it is not certain whether these genes play critical roles in the TET1-mediated MASLD progression.

Comment #6. Most regulated genes should be presented (perhaps a volcano plot).

Response: We followed the reviewer's suggestion to present the most regulated genes using a volcano plot. We also labeled the most altered genes (Top 50) on this volcano plot (Fig. 3F).

Comment #7. Since the authors suggest that Tet1 mediated regulation of CD36 gene expression is involved in fatty acid uptake and MASLD, it would be great to specifically investigate the effect of Tet1 on transcription factors that act as fatty acid sensors. In my opinion, there are two of these: HNF4alpha (simcox et al., Cell metab, 2017) and PPARalpha (fougerat et al., Cell rep, 2022).

Response: Thank you so much for the reviewer's comment about these two references. As we suggested that TET1 modulates MASLD through upregulating CD36, liver specific knockout of TET1 should alleviate MASLD via disrupting free acid uptakes in the liver by suppressing CD36 expression. Based on these two references, we measured Hnf4a, Cpt1a, Cpt1b, Octn2, and CrAT in control and TET1 LKO mice. We found downregulation of several Hnf4a downstream target genes, including Cpt1a, Octn2, and CrAT (SFig. 9B). Besides, we investigated Ppara, Fgf21, Cyp4a10, Cyp4a14, and Ehhadh in these mice. Several Ppara downstream targets, which include Fgf21, Cyp4a14 and Ehhadh, were found downregulated in TET1 LKO mice (SFig .9A). Together, these data support our findings that liver TET1 promotes MASLD progression through upregulating CD36 expression.

7th Feb 2025

Dear Dr. Huang,

Thank you for submitting your revised study. We have now received the reports from the referees who evaluated your revised manuscript. As you will see from the reports below, they are satisfied with the revisions, and I will therefore be able to accept your manuscript once the following minor editorial issues are addressed:

1/ Manuscript text:

- Please remove the highlighted yellow font and only keep in track changes mode any new modification.
- There is a discrepancy between Huiyi Lin (submission system) and Hui-Yi Lin (manuscript), please adjust.
- Please correct the order of the manuscript sections as follows: Abstract, Keywords, Introduction, Results, Discussion, Methods, Acknowledgements, Disclosure and competing interests statement, References, Figure legends, Expanded View Figure legends

- Methods:

- o "Materials and Methods" should be renamed "Methods".
- o The reagents and tools table should be removed from the manuscript text file.
- o The reference to BioRender should be removed from the reagents and tools table and added to the Methods as a section as outlined here:

Graphics:

(some of the... OR Figure #... OR synopsis) Graphics were created with BioRender.com.

Notes: "Materials and Methods" should be "Methods".

- o Cells: please indicate whether the cells were tested for mycoplasma contamination and authenticated.
- o Antibodies: please provide dilutions/concentrations for all.
- o Statistical analysis: please provide a statement on sample size, exclusion/inclusion criteria, blinding and randomization.
- Data Availability: Note that the Data Availability Section is restricted to new primary data that are part of this study (no previously published datasets). Please note that the data must be public before acceptance. Please remove "The datasets produced in this study are publicly available on the EMBO Molecular Medicine website."
- Author contributions: CRediT has replaced the traditional author contributions section because it offers a systematic machine readable author contributions format that allows for more effective research assessment. Please remove the Authors Contributions from the manuscript and use the free text boxes beneath each contributing author's name in our system to add specific details on the author's contribution. More information is available in our guide to authors.
- Please rename "Declaration of interests" to "Disclosure statement and competing interests".

2/ Figures and Appendix:

- Tables EV1-4 should be renamed Dataset EV1-4. Each excel file will need a legend added in a separate tab/worksheet. Table EV5 should be renumbered accordingly.
- We would suggest to make some of the Appendix figures EV figures. Indeed, we replaced Supplementary Information with Expanded View (EV) Figures and Tables that are collapsible/expandable online. A maximum of 5 EV Figures can be typeset. EV Figures should be cited as "Figure EV1, Figure EV2" etc... in the text and their respective legends should be included in the main text after the legends of regular figures.
- For the figures that you do NOT wish to display as Expanded View figures, they should be bundled together with their legends in a single PDF file called *Appendix*, which should start with a short Table of Content. Appendix figures should be referred to in the main text as: "Appendix Figure S1, Appendix Figure S2" etc.
- Additional Tables/Datasets should be labeled and referred to as Table EV1, Dataset EV1, etc. Legends have to be provided in a separate tab in case of .xls files. Alternatively, the legend can be supplied as a separate text file (README) and zipped together with the Table/Dataset file.

- Please make sure that all figures and figure panels are referenced in the text (currently, a callout is missing for Fig. 3I).
- Appendix: please correct the nomenclature to "Appendix Figure S1" etc. throughout; page numbers should be added to the table of contents.
- Please address the queries from our copy editors in the figure legends:
 1. Please note that the exact p values are not provided in the legends of figures 1A, E, F; 2A, C, D, E, H, I; 3B, C, D; 4A, B, C, G, H, J, L, N, Q; 5E-G; 6A, C, D, E, H, I, K, L; 7B-D, I, K, L, N
 2. Please indicate the statistical test used for data analysis in the legend of figure 3F.
 3. Please note that information related to n is missing in the legends of figures 1E, F; 2A, C, D, E, F, H, I; 3B, C, D, E, F; 4A, B, C, G, H, J, L, N, Q; 5E-G; 6A, C, D, E, G, H, I-L; 7B-F, H-L, N.
 4. Please note that the error bars are not defined in the legends of figures 4G, H.
 5. Please note that scale bar and its definition are missing for figures 4I, 6F, 7G.

3/ Source Data:

- Please carefully check the source data provided for Fig. 5E CD36-E2 pcDNA3 vs. pcDNA3-TET1.
- Please check the labeling of source data Fig. 7M.

4/ Checklist:

- Cell materials: please fill in the subsection on mycoplasma and authentication.
- Study design and statistics: please fill in all subsections.
- Data availability: please fill in the subsection "Primary datasets and Data Availability".

5/ Please provide "The paper explained": EMBO Molecular Medicine articles are accompanied by a summary of the articles to emphasize the major findings in the paper and their medical implications for the non-specialist reader. Please provide a draft summary of your article highlighting

6/ Synopsis:

- Thank you for providing a nice visual abstract. Please resize to 550 px wide x 300-600 px high and make sure that the text remains legible. A cropped portion of this image will serve as thumbnail for the table of content on our webpage.
- Please also provide a synopsis text: Synopses are displayed on the journal webpage and are freely accessible to all readers. They include a short stand first (maximum of 300 characters, including space) as well as 2-5 one-sentence bullet points that summarize the paper (maximum of 30 words / bullet point). Please write the bullet points to summarize the key new findings. They should be designed to be complementary to the abstract - i.e. not repeat the same text.

7/ As part of the EMBO Publications transparent editorial process initiative (see our Editorial at <http://embomolmed.embopress.org/content/2/9/329>), EMBO Molecular Medicine will publish online a Review Process File (RPF) to accompany accepted manuscripts.

This file will be published in conjunction with your paper and will include the anonymous referee reports, your point-by-point response and all pertinent correspondence relating to the manuscript. Let us know whether you agree with the publication of the RPF.

I look forward to receiving your revised manuscript.

Yours sincerely,

Lise Roth

***** Reviewer's comments *****

Referee #2 (Comments on Novelty/Model System for Author):

The authors have responded convincingly to my comments and have adjusted the manuscript accordingly. In particular, the addition of lipidomic analysis is a major plus.

Referee #3 (Remarks for Author):

The authors have improved the manuscript and addressed my concerns.

All editorial and formatting issues were resolved by the authors.

12th Mar 2025

Dear Dr. Huang,

Thank you for sending your revised files. I am pleased to inform you that your manuscript is accepted for publication and is now being sent to our publisher to be included in the next available issue of EMBO Molecular Medicine!

Yours sincerely,

Lise Roth
